# NEARLY $d$-LINEAR CONVERGENCE BOUNDS FOR DIFFUSION MODELS VIA STOCHASTIC LOCALIZATION

**Joe Benton**,* **Valentin De Bortoli**,† **Arnaud Doucet**,* **George Deligiannidis**\*

## ABSTRACT

Denoising diffusions are a powerful method to generate approximate samples from high-dimensional data distributions. Recent results provide polynomial bounds on their convergence rate, assuming $L^2$-accurate scores. Until now, the tightest bounds were either superlinear in the data dimension or required strong smoothness assumptions. We provide the first convergence bounds which are linear in the data dimension (up to logarithmic factors) assuming only finite second moments of the data distribution. We show that diffusion models require at most $\tilde{O}(\frac{d \log^2(1/\delta)}{\varepsilon^2})$ steps to approximate an arbitrary distribution on $\mathbb{R}^d$ corrupted with Gaussian noise of variance $\delta$ to within $\varepsilon^2$ in KL divergence. Our proof extends the Girsanov-based methods of previous works. We introduce a refined treatment of the error from discretizing the reverse SDE inspired by stochastic localization.

## 1 INTRODUCTION

Denoising diffusion models are a recent advance in generative modeling which have produced state-of-the-art results in many domains (Sohl-Dickstein et al., 2015; Song & Ermon, 2019; Ho et al., 2020; Song et al., 2021b), including image and text generation (Dhariwal & Nichol, 2021; Austin et al., 2021; Ramesh et al., 2022; Saharia et al., 2022), text-to-speech synthesis (Popov et al., 2021), and molecular structure modeling (Xu et al., 2022; Trippe et al., 2023; Watson et al., 2023). Denoising diffusion models take data samples, corrupt them through the iterated application of noise, and learn to reverse this noising procedure. For data on $\mathbb{R}^d$, we typically use a stochastic differential equation (SDE) as the noising process. Then, learning the reverse process is equivalent to learning the score of the noised distributions (Vincent, 2011; Song & Ermon, 2019; Song et al., 2021b).

Recently, significant progress has been made on improving our theoretical understanding of diffusion models, including several works which have established polynomial convergence bounds for such models (Chen et al., 2023a;d; Lee et al., 2023; Li et al., 2023). The current state-of-the-art bound without Lipschitz assumptions on the score of the data distribution is provided by Chen et al. (2023a), who show that diffusion models require at most $\tilde{O}(\frac{d^2 \log^2(1/\delta)}{\varepsilon^2})$ steps to approximate a given distribution to within $\varepsilon^2$ in KL divergence, assuming only a finite second moment of the target distribution and early stopping at time $\delta$. However, Chen et al. (2023a;d) suggest that the iteration complexity ought to scale linearly in the data dimension (see e.g. Chen et al. (2023d, Theorem 6)).

In this work, we close this gap. We derive the first convergence bounds for diffusion models which are linear in the data dimension (up to logarithmic factors) without smoothness assumptions. We deduce that an early stopped diffusion model has iteration complexity $\tilde{O}(\frac{d \log^2(1/\delta)}{\varepsilon^2})$. Our proof builds on Chen et al. (2023a;d), who use Girsanov's theorem to measure the distance between the true and approximate reverse paths. However, we provide a refined treatment of the time discretization error, using techniques from stochastic calculus to derive a differential inequality for the expected difference between the drift terms at different times. By bounding the terms of this differential inequality, we achieve tighter bounds on the difference between the true and approximate path measures.

A key ingredient in our proof will be Lemma 1, which allows us to perform several calculations explicitly. This result is inspired by stochastic localization (Eldan, 2013; 2020), developed to study the

---

*Department of Statistics, University of Oxford, {`benton,doucet,deligian`}`@stats.ox.ac.uk`
†CNRS, ENS Ulm, Paris, `valentin.debortoli@gmail.com`

KLS conjecture and applied to sampling. Stochastic localization sampling is equivalent to diffusion models (Montanari, 2023), letting us transfer insights from stochastic localization to our proof.

## 1.1 DIFFUSION MODELS

A diffusion model starts with a stochastic process $(X_t)_{t \in [0,T]}$ constructed by initializing $X_0$ in the data distribution $p_{\text{data}}$ and then evolving according to the Ornstein–Uhlenbeck (OU) SDE

$$\mathrm{d}X_t = -X_t \mathrm{d}t + \sqrt{2}\mathrm{d}B_t \quad \text{for } 0 \le t \le T, \tag{1}$$

where $(B_t)_{t \in [0,T]}$ is a Brownian motion on $\mathbb{R}^d$ (Song et al., 2021b). We then learn the dynamics of the reverse process $(Y_t)_{t \in [0,T]}$ defined by $Y_t = X_{T-t}$. If we let $q_t(\mathbf{x}_t)$ denote the marginals of the forward process, then under mild regularity conditions on $p_{\text{data}}$, the reverse process satisfies the SDE

$$\mathrm{d}Y_t = \{Y_t + 2\nabla \log q_{T-t}(Y_t)\}\mathrm{d}t + \sqrt{2}\mathrm{d}B'_t, \qquad Y_0 \sim q_T, \tag{2}$$

where $(B'_t)_{t \in [0,T]}$ is another Brownian motion (Anderson, 1982; Cattiaux et al., 2022). We can thus generate samples $\xi \sim p_{\text{data}}$ by sampling $Y_0 \sim q_T$, running the reverse SDE, and setting $\xi = Y_T$.

The OU process is a convenient choice of forward process as its transition densities are analytically tractable, with $q_{t|0}(\mathbf{x}_t|\mathbf{x}_0) = \mathcal{N}(\mathbf{x}_t; \mathbf{x}_0 e^{-t}, \sigma_t^2 I_d)$ where $\sigma_t^2 := 1 - e^{-2t}$. In what follows, we will denote the posterior mean $\mathbf{m}_t(\mathbf{x}_t) := \mathbb{E}_{q_{0|t}(\cdot|\mathbf{x}_t)}[X_0]$ and variance $\mathbf{\Sigma}_t(\mathbf{x}_t) := \text{Cov}_{q_{0|t}(\cdot|\mathbf{x}_t)}(X_0)$; $\mathbf{m}_t$ is related to the score function via $\nabla \log q_t(\mathbf{x}_t) = -\sigma_t^{-2}\mathbf{x}_t + e^{-t}\sigma_t^{-2}\mathbf{m}_t$ (see Lemma 5 below).

For convenience, we will assume throughout that our data distribution $p_{\text{data}}$ has identity covariance matrix (as is standard in applications), though our analysis holds similarly without this assumption. We also focus on an OU noising process rather than a general noising SDE (as in e.g. Chen et al. (2023e)), since the OU process is most common in practice. However, our results can be straight-forwardly extended to any linear SDE (including in particular the VE SDE (Song et al., 2021b)).

To simulate (2), we must make three approximations. First, as we do not have access to $\nabla \log q_t(\mathbf{x}_t)$ directly, we learn an approximation $s_\theta(\mathbf{x}_t, t)$ to $\nabla \log q_t(\mathbf{x}_t)$ for $t \in [0, T]$ by minimising

$$\mathcal{L}(s_\theta) = \int_0^T \mathbb{E}_{q_t}\left[\|s_\theta(X_t, t) - \nabla \log q_t(X_t)\|^2\right]\mathrm{d}t. \tag{3}$$

Although $\mathcal{L}(s_\theta)$ cannot be estimated directly, there are techniques such as denoising or implicit score matching which provide equivalent tractable objectives (Hyvärinen, 2005; Vincent, 2011). In practice, we empirically estimate these objectives using samples drawn from the forward process, which can be analytically simulated. We typically parameterize $s_\theta(\mathbf{x}_t, t)$ via a neural network for a vector of parameters $\theta \in \mathbb{R}^D$ and minimise the objective function using stochastic gradient descent.

Second, since we do not have access to the initial distribution $q_T$ of the reverse process, we instead initialize the reverse process in the standard Gaussian, which we denote $\pi_d$. This is a reasonable approximation since the OU process converges exponentially quickly to $\pi_d$ (Bakry et al., 2014).

Third, since (2) is a continuous-time process we must discretize time in order to simulate it. We pick time steps $0 = t_0 < t_1 < \cdots < t_N \le T$, sample $\hat{Y}_0 \sim \pi_d$ and define $(\hat{Y}_t)_{t \in [0,T]}$ via the SDE

$$\mathrm{d}\hat{Y}_t = \{\hat{Y}_t + 2s_\theta(\hat{Y}_{t_k}, T - t_k)\}\mathrm{d}t + \mathrm{d}\hat{B}_t \tag{4}$$

for each interval $[t_k, t_{k+1}]$ and $k = 0, \ldots, N-1$, for a Brownian motion $(\hat{B}_t)_{t \in [0,T]}$. We use the notation $\gamma_k := t_{k+1} - t_k$ for the step size and denote the marginals of the approximate reverse process by $p_t(\mathbf{x})$. The sampling scheme defined via (4) is known as the exponential integrator (Zhang & Chen, 2023; De Bortoli, 2022; Chen et al., 2023a). An alternative is the Euler–Maruyama (EM) scheme, which replaces the time-dependent drift term in (4) with its value at the start of the associated interval (Song et al., 2021b; Chen et al., 2023a). Both methods give similar asymptotic convergence rates, as indicated by Chen et al. (2023a), so we focus on the exponential integrator.

Finally, instead of running (4) all the way back to the start time, we perform early stopping and set $t_N = T - \delta$ for some small $\delta$. We do this because for non-smooth data distributions $\nabla \log q_t$ can blow up as $t \to 0$. This means that our model will approximate $q_\delta$ rather than $q_0 = p_{\text{data}}$, which is acceptable since for small $\delta$ the distance (e.g. in Wasserstein-$p$ metric) between $q_\delta$ and $p_{\text{data}}$ is small. Early stopping is frequently used in practical applications of diffusion models (Song et al., 2021b).

## 1.2 STOCHASTIC LOCALIZATION

Stochastic localization sampling schemes were developed by El Alaoui et al. (2022); Montanari (2023). To sample from a distribution $p_{\text{data}}$, we construct a measure-valued stochastic process $(\mu_s)_{s \geq 0}$ such that $\mu_s$ "localizes" as $s \to \infty$, meaning that there a.s. exists some $\xi$ such that $\mu_s \to \delta_\xi$ as $s \to \infty$, and $\xi \sim p_{\text{data}}$. We construct this process by sampling $\xi \sim p_{\text{data}}$ and defining a sequence $(U_s)_{s \geq 0}$ of noisy observations of $\xi$ via

$$U_s = s\xi + W_s, \tag{5}$$

where $(W_s)_{s \geq 0}$ is a Brownian motion on $\mathbb{R}^d$ (Montanari, 2023). We then define $\mu_s = \text{Law}(\xi \mid U_s)$, so that $\mu_s$ is a random measure depending on $U_s$. Since $U_s/s \to \xi$ almost surely as $s \to \infty$, we see $(\mu_s)_{s \geq 0}$ does indeed localize and $\lim_{s \to \infty} U_s/s$ is distributed according to $p_{\text{data}}$.

However, constructing $(U_s)_{s \geq 0}$ via (5) requires sampling $\xi \sim p_{\text{data}}$, which we cannot do. Fortunately, we avoid this using the observation that if $p_{\text{data}}$ has finite second moments, then $(U_s)_{s \geq 0}$ is equivalent in law to the unique solution to the SDE

$$dU_s = \mathbf{a}_s(U_s)ds + dW'_s, \tag{6}$$

where $(W'_s)_{s \geq 0}$ is a Brownian motion and $\mathbf{a}_s(U_s) = \mathbb{E}_{\mu_s}[\xi] = \mathbb{E}[\xi \mid U_s]$ (Liptser & Shiryaev, 1977; Montanari, 2023). This allows us to construct the stochastic localization process without direct access to $p_{\text{data}}$, so long as we have access to the function $\mathbf{a}_s(U_s)$. We also define $\mathbf{A}_s(U_s) = \text{Cov}(\mu_s)$.

Diffusion models and stochastic localization are equivalent under a time change (Montanari, 2023). If we define $(X_t)_{t \geq 0}$, $(U_s)_{s \geq 0}$ according to (1), (5) respectively and let $t(s) := \frac{1}{2}\log(1 + s^{-1})$, then $(U_s)_{s \geq 0}$ and $(se^{t(s)}X_{t(s)})_{s \geq 0}$ have the same law. In addition, $\mathbf{a}_s(U_s)$ and $\mathbf{m}_t(X_t)$ have the same law and $\mathbf{A}_s(U_s)$ and $\mathbf{\Sigma}_t(X_t)$ have the same law when $t = t(s)$. We will sometimes suppress the dependence of $\mathbf{a}_s, \mathbf{A}_s$ and $\mathbf{m}_t, \mathbf{\Sigma}_t$ on $U_s$ and $X_t$ respectively when the meaning is clear. For more details and an explicit derivation of the equivalence, see Appendix A.

We now recall two lemmas from the stochastic localization literature. The first can be found in Eldan (2013); Alaoui & Montanari (2022). The second follows from the first plus the argument in Eldan (2020). We provide proofs for both results in Appendix B for the reader's convenience.

**Proposition 1** (Alaoui & Montanari (2022), Theorem 2). *If we define* $L_s(\mathbf{x}) = \frac{d\mu_s}{dp_{\text{data}}}(\mathbf{x})$, *then* $dL_s(\mathbf{x}) = L_s(\mathbf{x})(\mathbf{x} - \mathbf{a}_s) \cdot dW'_s$ *for all* $s \geq 0$.

**Proposition 2** (Eldan (2020), Equation 11). *For all* $s \geq 0$, $\frac{d}{ds}\mathbb{E}[\mathbf{A}_s] = -\mathbb{E}[\mathbf{A}_s^2]$.

Translating Proposition 2 into the language of diffusion models, using that $\mathbf{A}_s$ and $\mathbf{\Sigma}_t$ are equal in law when $t = \frac{1}{2}\log(1 + s^{-1})$, we get the following directly from Proposition 2 and the chain rule.

**Lemma 1.** *For all* $t > 0$, $\frac{\sigma_t^3}{2\dot{\sigma}_t}\frac{d}{dt}\mathbb{E}[\mathbf{\Sigma}_t] = \mathbb{E}[\mathbf{\Sigma}_t^2]$.

Lemma 1 is the key insight that allows us to control the discretization error more precisely than previous works. In Lemma 5, we will see that $\mathbf{\Sigma}_t$ is related to the Jacobian of the score. Control of $\mathbf{\Sigma}_t$ via Lemma 1 will thus allow us to control time-discretization terms $E_{s,t}$ (defined in Section 3).

## 1.3 RELATED WORK

**Convergence of diffusion models** Initial results on the convergence of diffusion models required restrictive assumptions on the data distribution such as a log-Sobolev inequality (Lee et al., 2022; Yang & Wibisono, 2022), or produced bounds that were non-quantitative (Pidstrigach, 2022; Liu et al., 2022) or exponential in the problem parameters (De Bortoli et al., 2021; De Bortoli, 2022; Block et al., 2022). Recently however, several works have proven polynomial convergence rates for diffusion models (Chen et al., 2023a;d; Lee et al., 2023; Li et al., 2023).

First, Chen et al. (2023d) gave polynomial TV error bounds, assuming a Lipschitz score for all $t \geq 0$. They used Girsanov's theorem to measure the KL divergence between the true and approximate reverse processes (similar to Song et al. (2021a)), with an approximation argument since the standard assumptions for Girsanov's theorem do not hold in their setting. Second, Chen et al. (2023a) developed this method, introducing a tighter bound on the drift terms and an exponentially

| Regularity condition | Metric | Complexity | Result |
|---|---|---|---|
| $\forall t, \nabla \log q_t$ $L$-Lipschitz | $\mathrm{TV}(q_0, p_T)^2$ | $\tilde{O}\left(\frac{dL^2}{\varepsilon^2}\right)$ | (Chen et al., 2023d, Thm 2) |
| $\forall t, \nabla \log q_t$ $L$-Lipschitz | $\mathrm{KL}(q_0 \| p_T)$ | $\tilde{O}\left(\frac{dL^2}{\varepsilon^2}\right)$ | (Chen et al., 2023a, Thm 1) |
| None | $\mathrm{KL}(q_\delta \| p_{t_N})$ | $\tilde{O}\left(\frac{d^2 \log^2(1/\delta)}{\varepsilon^2}\right)$ | (Chen et al., 2023a, Thm 2) |
| None | $\mathrm{KL}(q_\delta \| p_{t_N})$ | $\tilde{O}\left(\frac{d \log^2(1/\delta)}{\varepsilon^2}\right)$ | This work: Corollary 1 |

Table 1: Summary of previous bounds and our results. Bounds expressed in terms of the number of steps required to guarantee an error of at most $\varepsilon^2$ in the stated metric, assuming perfect score estimation. We assume $p_{\text{data}}$ has finite second moments and is normalized so that $\mathrm{Cov}(p_{\text{data}}) = I_d$.

decaying sequence of time steps. They provide two bounds on the KL error. The first (Theorem 1) is linear in the data dimension $d$ but requires Lipschitz scores for $t \geq 0$ and depends quadratically on the Lipschitz constant. This is unideal since the Lipschitz assumption excludes many distributions of interest, such as those supported on a submanifold. In addition, the Lipschitz constant can hide additional dimension dependence in some cases (such as when the data are approximately supported on a submanifold). The second (Theorem 2) uses early stopping and applies to any data distribution with finite second moments but is quadratic in $d$. The latter gives the current best bound on the iteration complexity of diffusion models without smoothness assumptions of $\tilde{O}\left(\frac{d^2 \log^2(1/\delta)}{\varepsilon^2}\right)$[1].

In parallel, Lee et al. (2023) derived weaker polynomial convergence bounds using a $\chi^2$-based analysis and a method to convert $L^\infty$-accurate score estimates into $L^2$-accurate estimates. Most recently, Li et al. (2023) demonstrated bounds for several deterministic and non-deterministic discrete-time sampling methods using elementary techniques, assuming control of the approximate score and its derivative. We summarise the results of Chen et al. (2023a;d) and compare them to ours in Table 1.

In addition, several works have studied the convergence properties of deterministic or approximately deterministic sampling schemes based on diffusion models (Chen et al., 2023c;e; Albergo & Vanden-Eijnden, 2023; Albergo et al., 2023; Benton et al., 2023; Li et al., 2023). Other work has focused on the problem of score estimation. In particular, Oko et al. (2023) bound the error when approximating the score with a neural network and show that diffusion models are approximately minimax optimal for a certain class of target distributions, while Chen et al. (2023b) study the sample complexity and convergence properties of diffusion models when the data lies on a linear submanifold.

**Stochastic localization**  Stochastic localization was developed by Ronen Eldan to study isoperimetric inequalities (Eldan, 2013) such as the Kannan–Lovász–Simonovits (KLS) conjecture (Kannan et al., 1995) and the thin shell conjecture (Anttila et al., 2003; Bobkov & Koldobsky, 2003). It was based on the original localization methodologies of Lovász, Simonovits and Kannan (Lovász & Simonovits, 1993; Kannan et al., 1995) used to study high-dimensional and isoperimetric inequalities. Subsequently, stochastic localization has been used to make significant progress towards the KLS conjecture (Lee & Vempala, 2017; Chen, 2021), as well as to prove measure decomposition results (Eldan, 2020; Alaoui & Montanari, 2022) and for studying mixing times of Markov Chains (Chen & Eldan, 2022). Recently, several works developed algorithmic sampling techniques based on stochastic localization (El Alaoui et al., 2022; Montanari & Wu, 2023), and Montanari (2023) has shown that these sampling approaches are equivalent in the Gaussian setting to diffusion models.

**Concurrent work:**  After the first version of our work was made publicly available, an independent work appeared by Conforti et al. (2023) who derive similar bounds. In contrast to our work, they take a stochastic control perspective and work under a finite Fisher information smoothness condition. This condition can be removed with early stopping, giving bounds that are linear in data dimension up to logarithmic factors, similar to our own.

## 2  MAIN RESULTS

The core assumption required for our main result is the following control of the score approximation.

---

[1]Note that their bound is stated incorrectly in their abstract; the correct bound can be found in their Table 1.

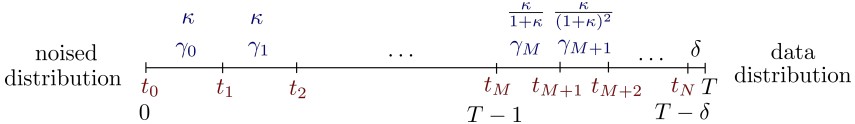

Figure 1: Illustration of a typical choice of step sizes satisfying $\gamma_k \leq \kappa \min\{1, T - t_{k+1}\}$.

**Assumption 1.** *The score approximation function $s_\theta(\mathbf{x}, t)$ satisfies*

$$\sum_{k=0}^{N-1} \gamma_k \mathbb{E}_{q_{t_k}} \left[ \|\nabla \log q_{T-t_k}(X_{t_k}) - s_\theta(X_{t_k}, T - t_k)\|^2 \right] \leq \varepsilon_{\text{score}}^2. \tag{7}$$

We can view (7) as a time-discretized version of $\mathcal{L}(s_\theta)$, so Assumption 1 suggests we learn a score approximation with $L^2$ error at most $\varepsilon_{\text{score}}$, adjusting for the discretization of the reverse process. Though this assumption is standard in the literature (Chen et al., 2023a;b), there may be statistical barriers to efficient estimation of the score in high dimensions (Biroli & Mézard, 2023; Ghio et al., 2023) and Assumption 1 may not capture the full practical difficulty of learning the score.

**Assumption 2.** *The data distribution $p_{\text{data}}$ has finite second moments, and $\text{Cov}(p_{\text{data}}) = I_d$.*

The first part of Assumption 2 is required for the convergence of the forward SDE. We include the second part for convenience when stating our results. Our analysis is not dependent on it, and we outline how to adapt our proofs for a general covariance matrix in Appendix C.

**Theorem 1.** *Suppose that Assumptions 1 and 2 hold, that $T \geq 1$, and that there is some $\kappa > 0$ such that for each $k = 0, \ldots, N - 1$ we have $\gamma_k \leq \kappa \min\{1, T - t_{k+1}\}$. Then,*

$$\text{KL}(q_\delta \| p_{t_N}) \lesssim \varepsilon_{\text{score}}^2 + \kappa^2 dN + \kappa dT + de^{-2T},$$

*where $f_1 \lesssim f_2$ denotes that there is a universal constant $C$ such that $f_1 \leq C f_2$.*

This is our main bound, and it consists of three parts. The first term $\varepsilon_{\text{score}}^2$ measures the error from using a learned rather than exact score. The second terms $\kappa^2 dN + \kappa dT$ are due to the discretization of the reverse SDE. The final term $de^{-2T}$ controls the convergence of the forward SDE.

We interpret $\kappa$ as controlling the maximum step size; $\gamma_k$ is bounded by $\kappa$ for $t \in [0, T-1]$, and for $t \in [T-1, T]$ the condition $\gamma_k \leq \kappa(T - t_{k+1})$ forces $\gamma_k$ to decay exponentially at rate $(1 + \kappa)^{-1}$. We visualise this in Figure 1. As in Corollary 1 below, for any $N$ there is a choice of time steps such that $\kappa = \tilde{O}(1/N)$, up to factors which are linear in $T$ and logarithmic in $1/\delta$. Since $T$ will scale logarithmically in $d$ and $\varepsilon_{\text{score}}$, we think of the second terms in Theorem 1 as scaling like $\tilde{O}(d/N)$.

We expect the first term $\varepsilon_{\text{score}}^2$ to scale linearly in $d$ in many cases, e.g. if the target distribution were the product of i.i.d. components. The convergence of the forward process is also linear in $d$. As such, Theorem 1 improves upon the previous state-of-the-art bounds (Chen et al., 2023a;d), which either require strong smoothness assumptions on $p_{\text{data}}$ or have at least quadratic dependence on $d$.

We next show that given $N$ we can choose a suitable sequence of time steps. This results in a bound on the iteration complexity of the diffusion model.

**Corollary 1.** *For $T \geq 1$, $\delta < 1$ and $N \geq \log(1/\delta)$, there exist $0 = t_0 < t_1 < \cdots < t_N = T - \delta$ such that for some $\kappa = \Theta\left(\frac{T + \log(1/\delta)}{N}\right)$ we have $\gamma_k \leq \kappa \min\{1, T - t_{k+1}\}$ for each $k = 0, \ldots, N-1$. Then, if we take $T = \frac{1}{2} \log\left(\frac{d}{\varepsilon_{\text{score}}^2}\right)$ and $N = \Theta\left(\frac{d(T + \log(1/\delta))^2}{\varepsilon_{\text{score}}^2}\right)$, we have $\text{KL}(q_\delta \| p_{t_N}) = O(\varepsilon_{\text{score}}^2)$. Hence, the diffusion model requires at most $\tilde{O}\left(\frac{d \log^2(1/\delta)}{\varepsilon^2}\right)$ steps to approximate $q_\delta$ to within $\varepsilon^2$ in KL divergence, assuming a sufficiently accurate score estimator.*

Corollary 1 shows that by making a suitable choice of $T$, $N$ and $t_0, \ldots, t_N$ we achieve a bound on the iteration complexity which is linear in the data dimension (up to logarithmic factors) under minimal smoothness assumptions, resolving the question raised in Chen et al. (2023a). The proof of Corollary 1 is deferred to Appendix D, where we show that taking half the time steps to be linearly spaced between 0 and $T - 1$ and half to be exponentially spaced between $T - 1$ and $T - \delta$, as illustrated in Figure 1 is sufficient. This choice is similar to that made in Chen et al. (2023a), reflecting their observation that exponentially decaying time steps appear optimal, at least theoretically.

The tightest previous bounds on the iteration complexity were obtained by Chen et al. (2023a;d). Assuming only finite second moments of $p_{\text{data}}$, Chen et al. (2023a) showed that diffusion models with early stopping have an iteration complexity of $\tilde{O}\left(\frac{d^2 \log^2(1/\delta)}{\varepsilon^2}\right)$. Alternatively, Chen et al. (2023a;d) showed that if the score is $L$-Lipschitz for all $t \geq 0$ then the iteration complexity is $\tilde{O}\left(\frac{dL^2}{\varepsilon^2}\right)$. The Lipschitz assumption can be removed using early stopping at the cost of an additional factor of $1/\delta^4$, arising from the fact that for an arbitrary data distribution the Lipschitz constant of $\nabla \log q_t$ explodes at rate $1/t^2$ as $t \to 0$ (Chen et al., 2023d, Lemma 20). As the distance between $q_0$ and $q_\delta$ scales with $d^{1/2}$ (e.g. in Wasserstein-$p$ metric), for a fixed Wasserstein-plus-KL (or Wasserstein-plus-TV) error we should scale $\delta$ proportionally to $d^{-1/2}$. As a result, for a fixed approximation error for $p_{\text{data}}$, the bounds of Chen et al. (2023d) and Chen et al. (2023a, Theorem 2) also scale superlinearly with $d$.

## 3 PROOF OF THEOREM 1

We now provide the proof of Theorem 1, which comes in three steps. We view Step 1 as our main novel contribution, while Steps 2 and 3 closely follow Chen et al. (2023a;d).

**Step 1:** We bound the error from discretizing the reverse SDE (see Lemma 2 below). Previous works bound $E_{s,t} := \mathbb{E}\left[\|\nabla \log q_{T-t}(Y_t) - \nabla \log q_{T-s}(Y_s)\|^2\right]$ using a Lipschitz assumption. We use a new Itô calculus argument to get a differential inequality for $E_{s,t}$ (Lemmas 3 and 4), relate the coefficients of this inequality to $\mathbf{m}_t$ and $\mathbf{\Sigma}_t$ using known results (Lemma 5), and bound the differential inequality coefficients using our key Lemma 1 (resulting in Lemmas 6 and 7).

**Step 2:** We bound the KL distance between the path measures of the true and approximate reverse process (Lemma 8). We use an analogous Girsanov-based method to Chen et al. (2023a;d).

**Step 3:** We use the data processing inequality to bound $\text{KL}(q_\delta \| p_{t_N})$ in terms of the distance between reverse path measures and the distance between $q_T$ and $\pi_d$. We bound the former using Step 2 and the latter using the convergence of the OU process (Proposition 4), as in Chen et al. (2023a;d).

Rather than work with two processes $(Y_t)_{t \in [0,T]}$ and $(\hat{Y}_t)_{t \in [0,T]}$ which are solutions to (2) and (4) under the same probability measure, it is more convenient to follow Chen et al. (2023a;d) and fix a single process $(Y_t)_{t \in [0,T]}$ and then define $Q$ and $P^{\pi_d}$ to be two different probability measures such that $(Y_t)_{t \in [0,T]}$ is a solution to (2) under $Q$ and to (4) under $P^{\pi_d}$. In addition, we define a probability measure $P^{q_T}$ under which $(Y_t)_{t \in [0,T]}$ is a solution to (4) but with the initial condition $Y_0 \sim q_T$.

### 3.1 BOUNDING THE DISCRETIZATION ERROR

**Lemma 2** (Bound on discretization error). *If $(Y_t)_{t \in [0,T]}$ is the solution to the SDE* (2), *then we have*

$$\sum_{k=0}^{N-1} \int_{t_k}^{t_{k+1}} \mathbb{E}_Q \left[\|\nabla \log q_{T-t}(Y_t) - \nabla \log q_{T-t_k}(Y_{t_k})\|^2\right] \mathrm{d}t \lesssim \kappa^2 dN + \kappa dT.$$

*Proof.* We start by controlling $E_{s,t}$ for $0 \leq s \leq t < T$, where $(Y_t)_{t \geq 0}$ follows the law $Q$ of the exact reverse process (2). Since $\nabla \log q_{T-t}(\mathbf{x})$ is smooth, we may apply Itô's lemma to get the following result, proved in Appendix E.

**Lemma 3.** *If $(Y_t)_{t \in [0,T]}$ is the solution to the SDE* (2), *then for all $t \in [0,T)$ we have*

$$\mathrm{d}(\nabla \log q_{T-t}(Y_t)) = -\nabla \log q_{T-t}(Y_t)\mathrm{d}t + \sqrt{2}\nabla^2 \log q_{T-t}(Y_t) \cdot \mathrm{d}B_t'.$$

From Lemma 3 and the product rule, we have $\mathrm{d}(e^t \nabla \log q_{T-t}(Y_t)) = \sqrt{2}e^t \nabla^2 \log q_{T-t}(Y_t) \cdot \mathrm{d}B_t'$. Since, by (13) below, $\int_s^t e^{2r} \mathbb{E}_Q\left[\|\nabla^2 \log q_{T-r}(Y_r)\|_F^2\right] \mathrm{d}r < \infty$ for $0 \leq s \leq t < T$, where we denote by $\|A\|_F = \text{Tr}(A^T A)^{1/2}$ the Frobenius norm of a matrix $A$, the RHS is a square-integrable martingale. So, we may apply Itô's isometry (Le Gall, 2016, Equation 5.8) and differentiate to get

$$\frac{\mathrm{d}}{\mathrm{d}t}\mathbb{E}_Q\left[\|e^t \nabla \log q_{T-t}(Y_t) - e^s \nabla \log q_{T-s}(Y_s)\|^2\right] = 2e^{2t}\mathbb{E}_Q\left[\|\nabla^2 \log q_{T-t}(Y_t)\|_F^2\right]. \quad (8)$$

In addition, for $0 \leq s \leq t < T$ where $s$ is considered fixed and $t$ may vary, from Lemma 3 we have

$$\mathrm{d}(\nabla \log q_{T-s}(Y_s) \cdot \nabla \log q_{T-t}(Y_t)) = -\nabla \log q_{T-s}(Y_s) \cdot \nabla \log q_{T-t}(Y_t)\mathrm{d}t$$
$$+ \sqrt{2}\nabla \log q_{T-s}(Y_s) \cdot \nabla^2 \log q_{T-t}(Y_t) \cdot \mathrm{d}B_t'.$$

Since $\int_s^t \mathbb{E}_Q \left[ \|\nabla^2 \log q_{T-r}(Y_r)\|_F^2 \right] \mathrm{d}r < \infty$ for $0 \leq s \leq t < T$, the final term is a square-integrable martingale. Integrating and taking expectations, we get

$$\frac{\mathrm{d}}{\mathrm{d}t}\mathbb{E}_Q \left[\nabla \log q_{T-s}(Y_s) \cdot \nabla \log q_{T-t}(Y_t)\right] = -\mathbb{E}_Q \left[\nabla \log q_{T-s}(Y_s) \cdot \nabla \log q_{T-t}(Y_t)\right], \quad (9)$$

where again we may interchange integration and expectation using Fubini. Combining (8) and (9), we deduce the following differential inequality for $E_{s,t}$. (See Appendix E for full derivation.)

**Lemma 4.** *For all $0 \leq s \leq t < T$, we have*

$$\frac{\mathrm{d}E_{s,t}}{\mathrm{d}t} = 2\mathbb{E}_Q \left[\|\nabla^2 \log q_{T-t}(Y_t)\|_F^2\right] - 2\mathbb{E}_Q \left[\|\nabla \log q_{T-t}(Y_t) - \nabla \log q_{T-s}(Y_s)\|^2\right]$$
$$+ 2\mathbb{E}_Q \left[\{\nabla \log q_{T-s}(Y_s) - \nabla \log q_{T-t}(Y_t)\} \cdot \nabla \log q_{T-s}(Y_s)\right]. \quad (10)$$

To further bound the RHS of (10), note that by Young's inequality

$$\mathbb{E}_Q \left[\{\nabla \log q_{T-s}(Y_s) - \nabla \log q_{T-t}(Y_t)\} \cdot \nabla \log q_{T-s}(Y_s)\right]$$
$$\leq \frac{1}{2}\left\{\mathbb{E}_Q \left[\|\nabla \log q_{T-t}(Y_t) - \nabla \log q_{T-s}(Y_s)\|^2\right] + \mathbb{E}_Q \left[\|\nabla \log q_{T-s}(Y_s)\|^2\right]\right\}.$$

Therefore,

$$\frac{\mathrm{d}E_{s,t}}{\mathrm{d}t} \leq \mathbb{E}_Q \left[\|\nabla \log q_{T-s}(Y_s)\|^2\right] + 2\mathbb{E}_Q \left[\|\nabla^2 \log q_{T-t}(Y_t)\|_F^2\right]. \quad (11)$$

We must thus bound $\mathbb{E}_Q \left[\|\nabla \log q_{T-s}(Y_s)\|^2\right]$ and $\mathbb{E}_Q \left[\|\nabla^2 \log q_{T-t}(Y_t)\|_F^2\right]$. We make use of the following lemma, which is found in previous work on diffusion models (see e.g. De Bortoli (2022); Lee et al. (2023); Benton et al. (2023)) and which we prove for completeness in Appendix E.

**Lemma 5.** *For all $t > 0$, we have $\nabla \log q_t(\mathbf{x}_t) = -\sigma_t^{-2}\mathbf{x}_t + e^{-t}\sigma_t^{-2}\mathbf{m}_t$ and $\nabla^2 \log q_t(\mathbf{x}_t) = -\sigma_t^{-2}I + e^{-2t}\sigma_t^{-4}\mathbf{\Sigma}_t$.*

We may use Lemma 5 to rewrite $\|\nabla \log q_{T-s}(Y_s)\|^2$ and $\|\nabla^2 \log q_{T-t}(Y_t)\|_F^2$ in terms of $Y_t$, $\mathbf{m}_t$ and $\mathbf{\Sigma}_t$. Expanding out the resulting expressions, the first can be bounded using elementary properties of the OU process, while the second can be bounded using properties of the OU process plus our key Lemma 1. We obtain the following inequalities (see Appendix E for derivations).

**Lemma 6.** *If $(Y_t)_{t \in [0,T]}$ is the solution to the reverse SDE (2), then for all $t, s \in [0, T]$ we have*

$$\mathbb{E}_Q \left[\|\nabla \log q_{T-s}(Y_s)\|^2\right] \leq d\sigma_{T-s}^{-2}, \quad (12)$$

*and*

$$\mathbb{E}_Q \left[\|\nabla^2 \log q_{T-t}(Y_t)\|_F^2\right] \leq d\sigma_{T-t}^{-4} - \frac{1}{2}\frac{\mathrm{d}}{\mathrm{d}r}\left(\sigma_{T-r}^{-4}\mathbb{E}\left[\mathrm{Tr}(\mathbf{\Sigma}_{T-r})\right]\right)|_{r=t}. \quad (13)$$

Putting the bounds in (12) and (13) together, we get

$$\mathbb{E}_Q \left[\|\nabla \log q_{T-s}(Y_s)\|^2\right] + 2\mathbb{E}_Q \left[\|\nabla^2 \log q_{T-t}(Y_t)\|_F^2\right]$$
$$\leq d\sigma_{T-s}^{-2} + 2d\sigma_{T-t}^{-4} - \frac{\mathrm{d}}{\mathrm{d}r}\left(\sigma_{T-r}^{-4}\mathbb{E}\left[\mathrm{Tr}(\mathbf{\Sigma}_{T-r})\right]\right)|_{r=t}.$$

Let us denote the two parts of this error term by

$$E_{s,t}^{(1)} := d\sigma_{T-s}^{-2} + 2d\sigma_{T-t}^{-4}, \qquad E_{s,t}^{(2)} := -\frac{\mathrm{d}}{\mathrm{d}r}\left(\sigma_{T-r}^{-4}\mathbb{E}\left[\mathrm{Tr}(\mathbf{\Sigma}_{T-r})\right]\right)|_{r=t}.$$

From (11), it follows that

$$E_{t_k,t} \leq \int_{t_k}^t \mathbb{E}_Q \left[\|\nabla \log q_{T-t_k}(Y_{t_k})\|^2\right] + 2\mathbb{E}_Q \left[\|\nabla^2 \log q_{T-s}(Y_s)\|_F^2\right] \mathrm{d}s \leq \int_{t_k}^t E_{t_k,s}^{(1)} + E_{t_k,s}^{(2)}\mathrm{d}s.$$
$$(14)$$

We now bound the contributions to $E_{t_k,t}$ from $E_{t_k,s}^{(1)}$ and $E_{t_k,s}^{(2)}$. It will be convenient to divide our time period into intervals $[0, T-1]$ and $[T-1, T-\delta]$ and treat these separately. We therefore assume there is an index $M$ with $t_M = T - 1$. This assumption is purely for presentation clarity and our argument works similarly without it. Then, the following lemma bounds the various contributions to $E_{t_k,t}$. The proof of Lemma 7 is elementary but technical, so we defer it to Appendix E.

**Lemma 7.** *The error terms $E_{t_k,s}^{(1)}$ satisfy*

$$\sum_{k=0}^{M-1} \int_{t_k}^{t_{k+1}} \left( \int_{t_k}^{t} E_{t_k,s}^{(1)} \mathrm{d}s \right) \mathrm{d}t \lesssim \kappa dT, \qquad \sum_{k=M}^{N-1} \int_{t_k}^{t_{k+1}} \left( \int_{t_k}^{t} E_{t_k,s}^{(1)} \mathrm{d}s \right) \mathrm{d}t \lesssim \kappa^2 dN, \qquad (15)$$

*and the error terms $E_{t_k,s}^{(2)}$ satisfy*

$$\sum_{k=0}^{N-1} \int_{t_k}^{t_{k+1}} \left( \int_{t_k}^{t} E_{t_k,s}^{(2)} \mathrm{d}s \right) \mathrm{d}t \lesssim \kappa d + \kappa^2 dN. \qquad (16)$$

Finally, by combining (14) with (15) and (16) we complete the proof of Lemma 2. $\qquad \square$

### 3.2 BOUNDING THE KL DISTANCE BETWEEN PATH MEASURES

**Lemma 8** (Bound on distance between path measures)**.** *If $Q$ and $P^{q_T}$ are the true and approximate path measures respectively then $\mathrm{KL}(Q||P^{q_T}) \lesssim \varepsilon_{\text{score}}^2 + \kappa^2 dN + \kappa dT$.*

*Proof.* We use the following result from Chen et al. (2023d) (proof recalled in Appendix F).

**Proposition 3** (Section 5.2 of Chen et al. (2023d))**.** *Let $Q$ and $P^{q_T}$ be the path measures of the solutions to (2) and (4) respectively, both started in $Y_0 \sim q_T$ and run from $t = 0$ to $t = t_N$. Assume that*

$$\sum_{k=0}^{N-1} \int_{t_k}^{t_{k+1}} \mathbb{E}_Q \left[ \|\nabla \log q_{T-t}(Y_t) - s_\theta(Y_{t_k}, T - t_k)\|^2 \right] \mathrm{d}t < \infty.$$

*Then, we have*

$$\mathrm{KL}(Q||P^{q_T}) \leq \sum_{k=0}^{N-1} \int_{t_k}^{t_{k+1}} \mathbb{E}_Q \left[ \|\nabla \log q_{T-t}(Y_t) - s_\theta(Y_{t_k}, T - t_k)\|^2 \right] \mathrm{d}t.$$

To apply Proposition 3, we note that by Lemma 2 and Assumption 1 we have

$$\sum_{k=0}^{N-1} \int_{t_k}^{t_{k+1}} \mathbb{E}_Q \left[ \|\nabla \log q_{T-t}(Y_t) - s_\theta(Y_{t_k}, T - t_k)\|^2 \right] \mathrm{d}t$$

$$\lesssim \sum_{k=0}^{N-1} \gamma_k \mathbb{E}_Q \left[ \|\nabla \log q_{T-t_k}(Y_{t_k}) - s_\theta(Y_{t_k}, T - t_k)\|^2 \right]$$

$$+ \sum_{k=0}^{N-1} \int_{t_k}^{t_{k+1}} \mathbb{E}_Q \left[ \|\nabla \log q_{T-t}(Y_t) - \nabla \log q_{T-t_k}(Y_{t_k})\|^2 \right] \mathrm{d}t$$

$$\lesssim \varepsilon_{\text{score}}^2 + \kappa^2 dN + \kappa dT < \infty.$$

Therefore, the conditions of Proposition 3 hold and Lemma 8 follows by applying Proposition 3. $\quad \square$

### 3.3 COMPLETING THE PROOF

Finally, we show how to complete the proof of Theorem 1 from Lemma 8. As a corollary of Lemma 8, we see that $Q$ is absolutely continuous with respect to $P^{q_T}$. Since $P^{q_T}$ and $P^{\pi_d}$ differ only by a change of starting distribution, we can write $\frac{\mathrm{d}P^{q_T}}{\mathrm{d}P^{\pi_d}}(\mathbf{y}) = \frac{\mathrm{d}q_T}{\mathrm{d}\pi_d}(\mathbf{y}_0)$ for a path $\mathbf{y} = (\mathbf{y}_t)_{t \in [0, t_N]}$,

and deduce that $P^{q_T}$ and $P^{\pi_d}$ are mutually absolutely continuous. It follows that $Q$ is absolutely continuous with respect to $P^{\pi_d}$ and $\frac{\mathrm{d}Q}{\mathrm{d}P^{\pi_d}}(\mathbf{y}) = \frac{\mathrm{d}Q}{\mathrm{d}P^{q_T}}(\mathbf{y})\frac{\mathrm{d}P^{q_T}}{\mathrm{d}P^{\pi_d}}(\mathbf{y}) = \frac{\mathrm{d}Q}{\mathrm{d}P^{q_T}}(\mathbf{y})\frac{\mathrm{d}q_T}{\mathrm{d}\pi_d}(\mathbf{y}_0)$. Therefore,

$$\mathrm{KL}(Q||P^{\pi_d}) = \mathbb{E}_Q\left[\log\left(\frac{\mathrm{d}Q}{\mathrm{d}P^{q_T}}(Y)\frac{\mathrm{d}q_T}{\mathrm{d}\pi_d}(Y_0)\right)\right] = \mathrm{KL}(Q||P^{q_T}) + \mathrm{KL}(q_T||\pi_d) \qquad (17)$$

The first term is bounded by Lemma 8. The second is controlled by the convergence of the forward process in KL divergence and can be bounded using the following proposition (which is very similar to Lemma 9 in Chen et al. (2023a)). We defer the proof of Proposition 4 to Appendix G.

**Proposition 4.** *Under Assumption 2, we have* $\mathrm{KL}(q_T||\pi_d) \lesssim de^{-2T}$ *for* $T \geq 1$.

Since $q_\delta$ and $p_{t_N}$ are the pushfowards of the path measures $Q$ and $P^{\pi_d}$ under $f : (\omega_t)_{t \in [0,t_N]} \mapsto \omega_{t_N}$, the data processing inequality implies that $\mathrm{KL}(q_\delta||p_{t_N}) \leq \mathrm{KL}(Q||P^{\pi_d})$. Finally, combining this with (17), Lemma 8, and Proposition 4 completes the proof of Theorem 1.

## 4 DISCUSSION

Inspecting our bound in Theorem 1, the error from approximating $q_T$ by $\pi_d$ decays exponentially in $T$ and so is typically negligible. If the $L^2$ error of our score approximation is $\varepsilon_{\mathrm{score}}^2$, then we cannot hope to improve on the term of order $\varepsilon_{\mathrm{score}}^2$ for the KL error due to using an approximate score. It remains to consider how tight the term corresponding to the discretization of the reverse process is.

Our proof shows that under our assumptions, with perfect score approximation and initializing the reverse SDE in $q_T$, the KL error induced by discretizing time is of order $\kappa^2 dN + \kappa dT$. As explained in Section 2, we can think of this as being $\tilde{O}(d/N)$, assuming a suitable choice of time steps and $\kappa$, or equivalently $\tilde{O}(d\eta)$ where $\eta$ is the average step size. We note that the linear dependence on $d$ (up to logarithmic factors) here is optimal (for justification, see Appendix H).

However, it is unclear whether the linear dependence on $\eta$ is optimal here. On the one hand, the KL divergence between the true and approximate reverse path measures is $\Theta(d\eta)$ in the worst case (consider the case where $p_{\mathrm{data}}$ is a point mass, or see Theorem 7 in Chen et al. (2023d) in the critically damped Langevin setting). Thus, our Girsanov-based method cannot improve upon the rate of $\tilde{O}(d\eta)$ without significant modification. In addition, the best known convergence rates in KL divergence for Langevin Monte Carlo (LMC) under various functional inequalities are $\tilde{O}(d\eta)$ (Cheng & Bartlett, 2018; Vempala & Wibisono, 2019; Chewi et al., 2022; Yang & Wibisono, 2022).

On the other hand, the increasing noise schedule of diffusion models may allow for improved convergence rates compared to LMC. As evidence, Mou et al. (2022) show that the EM discretization of an SDE has error $O(\eta^2)$ in reverse KL divergence, under smoothness assumptions which should be satisfied under early stopping, making only mild additional assumptions on the data distribution such as bounded support. We could apply their results to get $O(\eta^2)$ convergence bounds for the diffusion model, but the implicit constant would depend on $d$ and on the smoothness parameters, which in turn depend polynomially on $d$ and $1/\delta$. This leads to bounds which are quadratic in $\eta$ but superlinear in $d$. We expect such bounds to be tighter than our own when $\eta^{-1} \gg \mathrm{poly}(d)$.

We may also get better convergence bounds by working in weaker metrics. Under some smoothness assumptions on $p_{\mathrm{data}}$, our KL error bounds imply a bound of $\tilde{O}(\sqrt{\eta})$ in Wasserstein-2 distance via a Talagrand inequality (Otto & Villani, 2000). However, there is evidence that this rate is suboptimal. Under smoothness assumptions on the drift, the EM discretization has error $\tilde{O}(\eta)$ in Wasserstein-$\rho$ metric for $\rho \geq 1$ (Alfonsi et al., 2015). Under smoothness and convexity assumptions on the data distribution, LMC converges at rate $O(\eta)$ (Durmus & Moulines, 2019; Li et al., 2022). However, these results require additional smoothness assumptions and have superlinear dependence on $d$.

Ultimately, there appears to be a trade off between the dependence on the data dimension and the step size. Our key to obtaining bounds which are tight in $d$ was to use bounds on the drift coefficient of the reverse SDE which hold in expectation. All methods of which we are aware that achieve a better dependence on $\eta$ require stronger (e.g. $L^\infty$) control on the drift. These necessitate worse dimension dependence and additional smoothness assumptions. We leave bridging the gap between these two strands of proofs to future work.

ACKNOWLEDGMENTS

We thank Tyler Farghly for pointing out a small gap in the original version of this work. Joe Benton was supported by the EPSRC through the StatML CDT (EP/S023151/1). Arnaud Doucet acknowledges support from EPSRC grants EP/R034710/1 and EP/R018561/1.

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

## A Equivalence of Diffusion Models and Stochastic Localization

Suppose that $(X_t)_{t\geq 0}$ follows the OU SDE defined in (1). Using the integration by parts formula for continuous semimartingales (Le Gall, 2016),

$$\mathrm{d}(e^t X_t) = e^t X_t \mathrm{d}t + e^t\{-X_t\mathrm{d}t + \sqrt{2}\mathrm{d}B_t\} = \sqrt{2}e^t\mathrm{d}B_t.$$

By the Dubins–Schwarz theorem (Le Gall, 2016, Theorem 5.13), there is a process $(\hat{W}_s)_{s\geq 0}$ such that

$$\hat{W}_{e^{2s}-1} = \int_0^s \sqrt{2}e^r\mathrm{d}B_r$$

and $(\hat{W}_s)_{s\geq 0}$ is a standard Brownian motion with respect to the filtration $(\mathcal{F}_{\tau(s)})_{s\geq 0}$ where $\tau(s) = \frac{1}{2}\log(1+s)$. Then, for all $s\in(0,\infty)$ we can write

$$e^{\tau(s)}X_{\tau(s)} = X_0 + \hat{W}_s.$$

If we set $U_0 = 0$ and

$$U_s := se^{\tau(1/s)}X_{\tau(1/s)} = sX_0 + s\hat{W}_{1/s}$$

for $s\in(0,\infty)$, then we observe that $(U_s)_{s\geq 0}$ satisfies the definition of the stochastic localization process in (5), since the law of $(s\hat{W}_{1/s})_{s\geq 0}$ is the same as the law of $(W_s)_{s\geq 0}$. Thus the forward diffusion process and the stochastic localization process are equivalent under the chance of time variables $t(s) = \frac{1}{2}\log(1+s^{-1})$.

In addition, we see that conditioning on $U_s$ is equivalent to conditioning on $X_{t(s)}$ and thus $\mu_s$ and $q_{0|t}(\cdot|X_t)$ define the same distributions when $t = t(s)$. It follows that $\mathbf{a}_s(U_s)$ and $\mathbf{m}_t(X_t)$ have the same law and $\mathbf{A}_s(U_s)$ and $\mathbf{\Sigma}_t(X_t)$ have the same law when $t = t(s)$.

## B Proofs of Stochastic Localization Results

Propositions 1 and 2 are well-known and we reproduce the proofs here for convenience. Proposition 1 can be found for example in Eldan (2013); Alaoui & Montanari (2022); El Alaoui et al. (2022) and our proof of Proposition 2 is based on the argument on pages 8–9 of Eldan (2020).

*Proof of Proposition 1.* From (5), we can deduce that

$$\mu_s(\mathrm{d}\mathbf{x}) = \frac{1}{Z_s}\exp\left\{\mathbf{x}\cdot U_s - \frac{s}{2}\|\mathbf{x}\|^2\right\}p_{\text{data}}(\mathrm{d}\mathbf{x}),$$

where $Z_s = \int_{\mathbb{R}^d}\exp\left\{\mathbf{x}\cdot U_s - \frac{s}{2}\|\mathbf{x}\|^2\right\}p_{\text{data}}(\mathrm{d}\mathbf{x})$ is the normalizing constant. Therefore,

$$\mathrm{d}\log L_s(\mathbf{x}) = \mathbf{x}\cdot\mathrm{d}U_s - \frac{1}{2}\|\mathbf{x}\|^2\mathrm{d}s - \mathrm{d}\log Z_s. \tag{18}$$

Writing $h_s(\mathbf{x}) = \mathbf{x}\cdot U_s - \frac{s}{2}\|\mathbf{x}\|^2$ and using the definition of $U_s$ in (5) plus $\mathrm{d}[W,W]_s = \mathrm{d}s$, we have $\mathrm{d}h_s(\mathbf{x}) = \mathbf{x}\cdot\mathrm{d}U_s - \frac{1}{2}\|\mathbf{x}\|^2\mathrm{d}s$ and $\mathrm{d}[h(\mathbf{x}),h(\mathbf{x})]_s = \|\mathbf{x}\|^2\mathrm{d}s$. Since $h_s(\mathbf{x})$ is a continuous semi-martingale and $\exp$ is smooth, we may apply Itô's lemma to $\exp\{h_s(\mathbf{x})\}$ and integrate with respect to $p_{\text{data}}(\mathbf{x})$ to get

$$\mathrm{d}Z_s = \int_{\mathbb{R}^d}\left(\mathrm{d}h_s(\mathbf{x}) + \frac{1}{2}\mathrm{d}[h(\mathbf{x}),h(\mathbf{x})]_s\right)e^{h_s(\mathbf{x})}p_{\text{data}}(\mathrm{d}\mathbf{x})$$

$$= \int_{\mathbb{R}^d}\mathbf{x}\cdot\mathrm{d}U_s e^{h_s(\mathbf{x})}p_{\text{data}}(\mathrm{d}\mathbf{x})$$

$$= Z_s(\mathbf{a}_s\cdot\mathrm{d}U_s).$$

Then, via another application of Itô's lemma, since $Z_s$ is continuous and $\log$ is smooth,

$$\mathrm{d}\log Z_s = \frac{\mathrm{d}Z_s}{Z_s} - \frac{1}{2}\frac{d[Z]_s}{Z_s^2} = \mathbf{a}_s\cdot\mathrm{d}U_s - \frac{1}{2}\|\mathbf{a}_s\|^2\mathrm{d}s.$$

Substituting this into (18), we see that

$$d \log L_s(\mathbf{x}) = (\mathbf{x} - \mathbf{a}_s) \cdot (dU_s - \mathbf{a}_s ds) - \frac{1}{2}\|\mathbf{x} - \mathbf{a}_s\|^2 ds$$

$$= (\mathbf{x} - \mathbf{a}_s) \cdot dW_s' - \frac{1}{2}\|\mathbf{x} - \mathbf{a}_s\|^2 ds$$

where we recall the definition of $(W_s')_{s \geq 0}$ from (6). The result then follows via a final application of Itô's lemma. $\square$

*Proof of Proposition 2.* First, using Proposition 1 we have

$$d\mathbf{a}_s = d\left(\int_{\mathbb{R}^d} \mathbf{x} L_s(\mathbf{x}) p_{\text{data}}(d\mathbf{x})\right)$$

$$= \int_{\mathbb{R}^d} \mathbf{x} dL_s(\mathbf{x}) p_{\text{data}}(d\mathbf{x})$$

$$= \int_{\mathbb{R}^d} \mathbf{x} \otimes (\mathbf{x} - \mathbf{a}_s) L_s(\mathbf{x}) \cdot dW_s' \, p_{\text{data}}(d\mathbf{x}).$$

This implies that

$$d\mathbf{a}_s = \mathbb{E}_{\mu_s}\left[\xi \otimes (\xi - \mathbf{a}_s)\right] dW_s' = \mathbf{A}_s \cdot dW_s'.$$

It then follows from Itô's isometry that

$$\frac{d}{ds}\mathbb{E}\left[\mathbf{a}_s^{\otimes 2}\right] = \mathbb{E}\left[\mathbf{A}_s^2\right].$$

The result then follows from the fact that $\mathbb{E}\left[\mathbf{A}_s\right] = \mathbb{E}_{\mu_s}\left[\xi^{\otimes 2}\right] - \mathbb{E}\left[\mathbf{a}_s^{\otimes 2}\right]$. $\square$

## C ADAPTATIONS REQUIRED TO HANDLE A GENERAL COVARIANCE OF $p_{\text{data}}$

We briefly outline the changes required in our proofs to handle a data distribution with a general covariance matrix. The main changes required are in the proofs of Lemmas 6 and 7 and Proposition 4. In this section, we replace Assumption 2 with the following.

**Assumption 3.** *The data distribution $p_{\text{data}}$ has finite second moments, with $M_2 := \mathbb{E}_{p_{\text{data}}}\left[\|X_0\|^2\right]$.*

First, we note that the proofs of Lemmas 3, 4 and 5 go through unchanged, and so these results also hold in the more general setting. For Lemma 6, the proof of (12) can be adapted, replacing each time we use $\mathbb{E}\left[\|X_0\|^2\right] = d$ with $\mathbb{E}\left[\|X_0\|^2\right] = M_2$ and noting that since $X_0$ and $X_t - e^{-t}X_0$ are independent, we have $\mathbb{E}\left[\|X_t\|^2\right] = \mathbb{E}\left[\|(X_t - e^{-t}X_0) + e^{-t}X_0\|^2\right] = d\sigma_t^2 + e^{-2t}M_2$. We find that all instances of $M_2$ cancel in the final result and (12) holds unchanged. In addition, the proof of (13) needs no alteration.

The only change in the proof of Lemma 7 comes towards the end, when we assert that $\mathbb{E}\left[\text{Tr}(\mathbf{\Sigma}_t)\right] = \mathbb{E}\left[\mathbb{E}\left[\|X_0\|^2|X_t\right] - \|\mathbb{E}\left[X_0|X_t\right]\|^2\right] \leq \mathbb{E}\left[\|X_0\|^2\right] = d$ for $t \in [1, T]$. Under Assumption 3, this becomes $\mathbb{E}\left[\text{Tr}(\mathbf{\Sigma}_t)\right] \leq \mathbb{E}\left[\|X_0\|^2\right] = M_2$. Propagating this through the rest of the proof, we find that (16) should be replaced by

$$\sum_{k=0}^{N-1} \int_{t_k}^{t_{k+1}} \left(\int_{t_k}^{t} E_{t_k,s}^{(2)} ds\right) dt \lesssim \kappa M_2 + \kappa^2 dN. \tag{19}$$

The final change that must be made is to Proposition 4, which must be replaced by the following more general version that can be proved along identical lines.

**Proposition 5.** *Under Assumption 3, we have $\text{KL}(q_T || \pi_d) \lesssim (d + M_2)e^{-2T}$ for $T \geq 1$.*

Putting all of these changes together, we arrive at the following more general version of Theorem 1.

**Theorem 2.** *Suppose that Assumptions 1 and 3 hold, that $T \geq 1$, and that there is some $\kappa > 0$ such that for each $k = 0, \ldots, N - 1$ we have $\gamma_k \leq \kappa \min\{1, T - t_{k+1}\}$. Then,*

$$\text{KL}(q_\delta || p_{t_N}) \lesssim \varepsilon_{\text{score}}^2 + \kappa^2 dN + \kappa dT + \kappa M_2 + (d + M_2)e^{-2T}.$$

Theorem 2 holds even in the case where the covariance is not strictly positive definite, and in the case where it is unknown to our diffusion model algorithm. Since we expect $M_2$ to scale linearly in $d$ in most cases, we consider this result to be in essentially the same spirit as Theorem 1.

## D  PROOF OF COROLLARY 1

*Proof of Corollary 1.* For convenience, we assume that $N$ is even. We take $t_0 = 0$, $t_{N/2} = T - 1$, and $t_N = T - \delta$, and pick $t_1, \ldots, t_{N-1}$ such that $t_0, \ldots, t_{N/2}$ are linearly spaced on $[0, T - 1]$ and $T - t_{N/2}, \ldots, T - t_N$ are an exponentially decaying sequence from 1 to $\delta$ (illustrated in Figure 1).

Then, $\gamma_k \leq \kappa \min\{1, T - t_{k+1}\}$ for each $k = 0, \ldots, N - 1$ if and only if $\kappa \geq (T - 1)/(N/2)$ and $\kappa \geq (1/\delta)^{1/N} - 1$. The former is satisfied if we take $\kappa = \Omega\left(\frac{T}{N}\right)$ and the second is satisfied provided that $\kappa = \Omega\left(\frac{\log(1/\delta)}{N}\right)$, since $N \geq \log(1/\delta)$ and $e^x \leq 1 + (e - 1)x$ for $x \leq 1$. Therefore, for some $\kappa = \Theta\left(\frac{T + \log(1/\delta)}{N}\right)$ we have $\gamma_k \leq \kappa \min\{1, T - t_{k+1}\}$ for each $k = 0, \ldots, N - 1$, proving the first part of Corollary 1.

For the second part, suppose that we set $T = \frac{1}{2}\log\left(\frac{d}{\varepsilon_{\text{score}}^2}\right)$ and $N = \Theta\left(\frac{d(T + \log(1/\delta))^2}{\varepsilon_{\text{score}}^2}\right)$. Then, we have $\kappa^2 dN = O(\varepsilon_{\text{score}}^2)$, $\kappa dT = O(\varepsilon_{\text{score}}^2)$, and $de^{-2T} = O(\varepsilon_{\text{score}}^2)$. We may therefore apply Theorem 1 to get that $\text{KL}(q_\delta \| p_{t_N}) = O(\varepsilon_{\text{score}}^2)$. The bound on the iteration complexity then follows since $T$ depends only logarithmically on $d$ and $\varepsilon_{\text{score}}$. $\square$

## E  OMITTED PROOFS FROM SECTION 3

Here, we provide the proofs of Lemmas 3, 4, 5, 6 and 7 which were omitted from Section 3.

*Proof of Lemma 3.* Recall that the reverse process $(Y_t)_{t \in [0,T]}$ satisfies

$$\mathrm{d}Y_t = \{Y_t + 2\nabla \log q_{T-t}(Y_t)\}\mathrm{d}t + \sqrt{2}\mathrm{d}B_t'.$$

Since $\nabla \log q_{T-t}(\mathbf{x})$ is smooth for $t \in [0, T)$, by Itô's lemma we can write

$$
\begin{aligned}
\mathrm{d}(\nabla \log q_{T-t}(Y_t)) = {} & \left\{\nabla^2 \log q_{T-t}(X_t) \cdot \{Y_t + 2\nabla \log q_{T-t}(Y_t)\} + \Delta(\nabla \log q_{T-t})(Y_t)\right\}\mathrm{d}t \\
& + \frac{\mathrm{d}(\nabla \log q_{T-t})(Y_t)}{\mathrm{d}t}\mathrm{d}t + \sqrt{2}\nabla^2 \log q_{T-t}(Y_t) \cdot \mathrm{d}B_t'. \quad (20)
\end{aligned}
$$

The Fokker–Planck equation for the forward process is

$$\mathrm{d}q_t(\mathbf{x}) = \{-\nabla \cdot (-\mathbf{x}q_t(\mathbf{x})) + \Delta q_t(\mathbf{x})\}\mathrm{d}t,$$

from which we can deduce

$$\mathrm{d}(\log q_t)(\mathbf{x}) = \{d + \mathbf{x} \cdot \nabla \log q_t(\mathbf{x}) + \Delta \log q_t(\mathbf{x}) + \|\nabla \log q_t(\mathbf{x})\|^2\}\mathrm{d}t.$$

It follows that

$$
\begin{aligned}
\frac{\mathrm{d}(\nabla \log q_{T-t})}{\mathrm{d}t}(\mathbf{x}) = {} & -\{\nabla \log q_{T-t}(\mathbf{x}) + \nabla^2 \log q_{T-t}(\mathbf{x}) \cdot \mathbf{x} + \nabla(\Delta \log q_{T-t}(\mathbf{x})) \\
& + 2\nabla^2 \log q_{T-t}(\mathbf{x}) \cdot \nabla \log q_{T-t}(\mathbf{x})\},
\end{aligned}
$$

by interchanging the order of the derivative operators. Substituting this into (20) and simplifying, we obtain the desired result. $\square$

*Proof of Lemma 4.* First, expanding (8) we get

$$
\begin{aligned}
& \frac{\mathrm{d}}{\mathrm{d}t}\mathbb{E}_Q\left[\|\nabla \log q_{T-t}(Y_t)\|^2\right] + 2\mathbb{E}_Q\left[\|\nabla \log q_{T-t}(Y_t)\|^2\right] \\
& \qquad\qquad - 2e^{-(t-s)}\frac{\mathrm{d}}{\mathrm{d}t}\mathbb{E}_Q\left[\nabla \log q_{T-s}(Y_s) \cdot \nabla \log q_{T-t}(Y_t)\right] \\
& \qquad\qquad\qquad - 2e^{-(t-s)}\mathbb{E}_Q\left[\nabla \log q_{T-s}(Y_s) \cdot \nabla \log q_{T-t}(Y_t)\right] \\
& = 2\mathbb{E}_Q\left[\|\nabla^2 \log q_{T-t}(Y_t)\|_F^2\right].
\end{aligned}
$$

Combined with (9), this shows that

$$\frac{\mathrm{d}}{\mathrm{d}t}\mathbb{E}_Q\left[\|\nabla \log q_{T-t}(Y_t)\|^2\right] + 2\mathbb{E}_Q\left[\|\nabla \log q_{T-t}(Y_t)\|^2\right] = 2\mathbb{E}_Q\left[\|\nabla^2 \log q_{T-t}(Y_t)\|_F^2\right].$$

Then,

$$
\begin{aligned}
\frac{\mathrm{d}E_{s,t}}{\mathrm{d}t} &= \frac{\mathrm{d}}{\mathrm{d}t}\mathbb{E}_Q\left[\|\nabla \log q_{T-t}(Y_t)\|^2\right] - 2\frac{\mathrm{d}}{\mathrm{d}t}\mathbb{E}_Q\left[\nabla \log q_{T-s}(Y_s)\cdot \nabla \log q_{T-t}(Y_t)\right] \\
&= 2\mathbb{E}_Q\left[\|\nabla^2 \log q_{T-t}(Y_t)\|_F^2\right] - 2\mathbb{E}_Q\left[\|\nabla \log q_{T-t}(Y_t)\|^2\right] \\
&\quad + 2\mathbb{E}_Q\left[\nabla \log q_{T-s}(Y_s)\cdot \nabla \log q_{T-t}(Y_t)\right]
\end{aligned}
$$

which rearranges to give (10). $\qquad\square$

*Proof of Lemma 5.* Part (i) is a classical result, sometimes known as Tweedie's formula (Robbins, 1956). Part (ii) has been established in previous works (see e.g. De Bortoli (2022, Lemma C.2) or Lee et al. (2023, Lemma 4.13)). We provide proofs of both results for reference.

For (i), we have

$$
\nabla \log q_t(\mathbf{x}_t) = \frac{1}{q_t(\mathbf{x}_t)}\int_{\mathbb{R}^d}\nabla \log q_{t|0}(\mathbf{x}_t|\mathbf{x}_0)q_{0,t}(\mathbf{x}_0,\mathbf{x}_t)\mathrm{d}\mathbf{x}_0.
$$

Since $q_{t|0}(\mathbf{x}_t|\mathbf{x}_0) = \mathcal{N}(\mathbf{x}_t;\mathbf{x}_0 e^{-t},\sigma_t^2 I)$, it follows that $\nabla \log q_{t|0}(\mathbf{x}_t|\mathbf{x}_0) = -\sigma_t^{-2}(\mathbf{x}_t - \mathbf{x}_0 e^{-t})$. Therefore,

$$
\begin{aligned}
\nabla \log q_t(\mathbf{x}_t) &= \mathbb{E}_{q_{0|t}(\cdot|\mathbf{x}_t)}\left[-\sigma_t^{-2}(\mathbf{x}_t - X_0 e^{-t})\right] \\
&= -\sigma_t^{-2}\mathbf{x}_t + e^{-t}\sigma_t^{-2}\mathbf{m}_t.
\end{aligned}
$$

For (ii), we can write

$$
\begin{aligned}
&\nabla^2 \log q_t(\mathbf{x}_t) \\
&= \frac{1}{q_t(\mathbf{x}_t)}\int_{\mathbb{R}^d}\nabla^2 \log q_{t|0}(\mathbf{x}_t|\mathbf{x}_0)q_{0,t}(\mathbf{x}_0,\mathbf{x}_t)\mathrm{d}\mathbf{x}_0 \\
&\quad + \frac{1}{q_t(\mathbf{x}_t)}\int_{\mathbb{R}^d}(\nabla \log q_{t|0}(\mathbf{x}_t|\mathbf{x}_0)(\nabla \log q_{t|0}(\mathbf{x}_t|\mathbf{x}_0))^T q_{0,t}(\mathbf{x}_0,\mathbf{x}_t)\mathrm{d}\mathbf{x}_0 \\
&\quad - \frac{1}{q_t(\mathbf{x}_t^2)}\left(\int_{\mathbb{R}^d}\nabla \log q_{t|0}(\mathbf{x}_t|\mathbf{x}_0)q_{0,t}(\mathbf{x}_0,\mathbf{x}_t)\mathrm{d}\mathbf{x}_0\right)\left(\int_{\mathbb{R}^d}\nabla \log q_{t|0}(\mathbf{x}_t|\mathbf{x}_0)q_{0,t}(\mathbf{x}_0,\mathbf{x}_t)\mathrm{d}\mathbf{x}_0\right)^T \\
&= -\frac{1}{\sigma_t^2}I + \mathbb{E}_{q_{0|t}(\cdot|\mathbf{x}_t)}\left[\sigma_t^{-4}(\mathbf{x}_t - X_0 e^{-t})(\mathbf{x}_t - X_0 e^{-t})^T\right] \\
&\quad - \mathbb{E}_{q_{0|t}(\cdot|\mathbf{x}_t)}\left[-\sigma_t^2(\mathbf{x}_t - X_0 e^{-t})\right]\mathbb{E}_{q_{0|t}(\cdot|\mathbf{x}_t)}\left[-\sigma_t^2(\mathbf{x}_t - X_0 e^{-t})\right]^T \\
&= -\sigma_t^{-2}I + \sigma_t^{-4}\mathrm{Cov}_{q_{0|t}(\cdot|\mathbf{x}_t)}(\mathbf{x}_t - X_0 e^{-t}) \\
&= -\sigma_t^{-2}I + e^{-2t}\sigma_t^{-4}\mathbf{\Sigma}_t.
\end{aligned}
$$

$\qquad\square$

*Proof of Lemma 6.* First, using the first part of Lemma 5 and expanding, we see that

$$
\mathbb{E}_{q_t}\left[\|\nabla \log q_t(X_t)\|^2\right] = \sigma_t^{-4}\mathbb{E}\left[\|X_t\|^2\right] - 2e^{-t}\sigma_t^{-4}\mathbb{E}\left[X_t \cdot \mathbf{m}_t\right] + e^{-2t}\sigma_t^{-4}\mathbb{E}\left[\|\mathbf{m}_t\|^2\right],
$$

where all expectations are with respect to $X_t \sim q_t$. Then,

$$
\mathbb{E}\left[X_t \cdot \mathbf{m}_t\right] = \mathbb{E}\left[X_t \cdot \mathbb{E}\left[X_0|X_t\right]\right] = \mathbb{E}\left[X_t \cdot X_0\right] = \mathbb{E}\left[X_0 \cdot \mathbb{E}\left[X_t|X_0\right]\right] = e^{-t}\mathbb{E}\left[\|X_0\|^2\right] = de^{-t}.
$$

Also, $\mathrm{Tr}(\mathbf{\Sigma}_t) = \mathbb{E}\left[\|X_0\|^2|\mathbf{x}_t\right] - \|\mathbf{m}_t\|^2$, so $\mathbb{E}\left[\|\mathbf{m}_t\|^2\right] = d - \mathbb{E}\left[\mathrm{Tr}(\mathbf{\Sigma}_t)\right]$. We conclude that

$$
\mathbb{E}_{q_t}\left[\|\nabla \log q_t(X_t)\|^2\right] = d\sigma_t^{-2} - \dot{\sigma}_t\sigma_t^{-3}\mathbb{E}\left[\mathrm{Tr}(\mathbf{\Sigma}_t)\right],
$$

where we have used $\dot{\sigma}_t\sigma_t = e^{-2t}$. This implies that

$$
\mathbb{E}_Q\left[\|\nabla \log q_{T-s}(Y_s)\|^2\right] \le d\sigma_{T-s}^{-2}.
$$

Second, using the second part of Lemma 5 along with the definition $\|A\|_F^2 = \mathrm{Tr}(A^T A)$ and expanding, we see that

$$
\mathbb{E}_{q_t}\left[\|\nabla^2 \log q_t(X_t)\|_F^2\right] = d\sigma_t^{-4} - 2\dot{\sigma}_t\sigma_t^{-5}\mathbb{E}\left[\mathrm{Tr}(\mathbf{\Sigma}_t)\right] + \dot{\sigma}_t^2\sigma_t^{-6}\mathbb{E}\left[\mathrm{Tr}(\mathbf{\Sigma}_t^2)\right].
$$

Taking traces in Lemma 1, we get

$$\frac{\sigma_t^4}{2e^{-2t}}\frac{\mathrm{d}}{\mathrm{d}t}\mathbb{E}\left[\mathrm{Tr}(\boldsymbol{\Sigma}_t)\right] = \frac{\sigma_t^4}{2\dot{\sigma}_t\sigma_t}\frac{\mathrm{d}}{\mathrm{d}t}\mathbb{E}\left[\mathrm{Tr}(\boldsymbol{\Sigma}_t)\right] = \frac{\sigma_t^3}{2\dot{\sigma}_t}\frac{\mathrm{d}}{\mathrm{d}t}\mathbb{E}\left[\mathrm{Tr}(\boldsymbol{\Sigma}_t)\right] = \mathbb{E}\left[\mathrm{Tr}(\boldsymbol{\Sigma}_t^2)\right],$$

and since $\mathbb{E}\left[\mathrm{Tr}(\boldsymbol{\Sigma}_t^2)\right] \geq 0$, it follows that $\frac{\mathrm{d}}{\mathrm{d}t}\mathbb{E}\left[\mathrm{Tr}(\boldsymbol{\Sigma}_t)\right] \geq 0$. Therefore,

$$\mathbb{E}_{q_t}\left[\|\nabla^2 \log q_t(X_t)\|_F^2\right] = d\sigma_t^{-4} - 2\dot{\sigma}_t\sigma_t^{-5}\mathbb{E}\left[\mathrm{Tr}(\boldsymbol{\Sigma}_t)\right] + \frac{1}{2}\dot{\sigma}_t\sigma_t^{-3}\frac{\mathrm{d}}{\mathrm{d}t}\mathbb{E}\left[\mathrm{Tr}(\boldsymbol{\Sigma}_t)\right]$$

$$\leq d\sigma_t^{-4} + \frac{1}{2}\frac{\mathrm{d}}{\mathrm{d}t}\left(\sigma_t^{-4}\mathbb{E}\left[\mathrm{Tr}(\boldsymbol{\Sigma}_t)\right]\right),$$

where we have used that $\sigma_t\dot{\sigma}_t \leq 1$. This implies that

$$\mathbb{E}_Q\left[\|\nabla^2 \log q_{T-t}(Y_t)\|_F^2\right] \leq d\sigma_{T-t}^{-4} + \frac{1}{2}\frac{\mathrm{d}}{\mathrm{d}r}\left(\sigma_r^{-4}\mathbb{E}\left[\mathrm{Tr}(\boldsymbol{\Sigma}_r)\right]\right)|_{r=T-t}$$

$$\leq d\sigma_{T-t}^{-4} - \frac{1}{2}\frac{\mathrm{d}}{\mathrm{d}r}\left(\sigma_{T-r}^{-4}\mathbb{E}\left[\mathrm{Tr}(\boldsymbol{\Sigma}_{T-r})\right]\right)|_{r=t}.$$

$\square$

*Proof of Lemma 7.* First we control the error terms $E_{t_k,s}^{(1)}$. If $s,t \in [0, T-1]$ then $\sigma_{T-s}^2, \sigma_{T-t}^2 \geq 1/2$ and so $E_{s,t}^{(1)} \leq 10d$. We therefore have

$$\sum_{k=0}^{M-1}\int_{t_k}^{t_{k+1}}\left(\int_{t_k}^t E_{t_k,s}^{(1)}\mathrm{d}s\right)\mathrm{d}t \leq 5d\sum_{k=0}^{M-1}\gamma_k^2$$

$$\lesssim \kappa dT,$$

since we have assumed that $\gamma_k \leq \kappa$. This proves the first part of (15).

If $s,t \in [T-1, T-\delta]$ then $(T-s)/2 \leq \sigma_{T-s}^2 \leq 2(T-s)$ and similarly for $t$. Therefore,

$$\sum_{k=M}^{N-1}\int_{t_k}^{t_{k+1}}\left(\int_{t_k}^t E_{t_k,s}^{(1)}\mathrm{d}s\right)\mathrm{d}t \leq 12d\sum_{k=M}^{N-1}\int_{t_k}^{t_{k+1}}\left(\int_{t_k}^t (T-s)^{-2}\mathrm{d}s\right)$$

$$\leq 12d\sum_{k=M}^{N-1}\frac{\gamma_k^2}{(T-t_{k+1})^2}$$

$$\lesssim \kappa^2 dN$$

since we have assumed that $\gamma_k \leq \kappa(T-t_{k+1})$. This proves the second part of (15).

Next, we control the error terms $E_{t_k,s}^{(2)}$. Note that $\sigma_{T-t}^{-4}$ is increasing in $t$ and $\mathbb{E}\left[\mathrm{Tr}(\Sigma_{T-t})\right]$ is decreasing in $t$ by Lemma 1. Therefore,

$$\sum_{k=0}^{N-1}\int_{t_k}^{t_{k+1}}\left(\int_{t_k}^t E_{t_k,s}^{(2)}\mathrm{d}s\right)\mathrm{d}t \leq \sum_{k=0}^{N-1}\int_{t_k}^{t_{k+1}}\left(\sigma_{T-t_k}^{-4}\mathbb{E}\left[\mathrm{Tr}(\boldsymbol{\Sigma}_{T-t_k})\right] - \sigma_{T-t}^{-4}\mathbb{E}\left[\mathrm{Tr}(\boldsymbol{\Sigma}_{T-t})\right]\right)\mathrm{d}t$$

$$\leq \sum_{k=0}^{N-1}\gamma_k\left(\sigma_{T-t_k}^{-4}\mathbb{E}\left[\mathrm{Tr}(\boldsymbol{\Sigma}_{T-t_k})\right] - \sigma_{T-t_k}^{-4}\mathbb{E}\left[\mathrm{Tr}(\boldsymbol{\Sigma}_{T-t_{k+1}})\right]\right).$$

We split this sum into $k = 0, \ldots, M-1$ and $k = M, \ldots, N-1$. For $k = 0, \ldots, M-1$, we have $\gamma_k\sigma_{T-t_k}^{-4} \leq 4\gamma_k \leq 4\kappa$ and so

$$\sum_{k=0}^{M-1}\gamma_k\left(\sigma_{T-t_k}^{-4}\mathbb{E}\left[\mathrm{Tr}(\boldsymbol{\Sigma}_{T-t_k})\right] - \sigma_{T-t_k}^{-4}\mathbb{E}\left[\mathrm{Tr}(\boldsymbol{\Sigma}_{T-t_{k+1}})\right]\right)$$

$$\leq 4\kappa\sum_{k=0}^{N-1}\left(\mathbb{E}\left[\mathrm{Tr}(\boldsymbol{\Sigma}_{T-t_k})\right] - \mathbb{E}\left[\mathrm{Tr}(\boldsymbol{\Sigma}_{T-t_{k+1}})\right]\right)$$

$$\leq 4\kappa\mathbb{E}\left[\mathrm{Tr}(\boldsymbol{\Sigma}_T)\right].$$

For $k = M, \ldots, N-1$ we have $\gamma_k \sigma_{T-t_k}^{-4} \leq 4\gamma_k/(T-t_k)^2 \leq 4\kappa/(T-t_k)$ and so

$$\sum_{k=M}^{N-1} \gamma_k \left( \sigma_{T-t_k}^{-4} \mathbb{E}\left[ \mathrm{Tr}(\boldsymbol{\Sigma}_{T-t_k}) \right] - \sigma_{T-t_k}^{-4} \mathbb{E}\left[ \mathrm{Tr}(\boldsymbol{\Sigma}_{T-t_{k+1}}) \right] \right)$$

$$\leq 4\kappa \sum_{k=M}^{N-1} \frac{1}{(T-t_k)} \left( \mathbb{E}\left[ \mathrm{Tr}(\boldsymbol{\Sigma}_{T-t_k}) \right] - \mathbb{E}\left[ \mathrm{Tr}(\boldsymbol{\Sigma}_{T-t_{k+1}}) \right] \right)$$

$$\leq 4\kappa \mathbb{E}\left[ \mathrm{Tr}(\boldsymbol{\Sigma}_1) \right] + 4\kappa \sum_{k=M+1}^{N-1} \frac{\gamma_{k-1}}{(T-t_k)(T-t_{k-1})} \mathbb{E}\left[ \mathrm{Tr}(\boldsymbol{\Sigma}_{T-t_k}) \right]$$

$$\leq 4\kappa \mathbb{E}\left[ \mathrm{Tr}(\boldsymbol{\Sigma}_1) \right] + 4\kappa^2 \sum_{k=M+1}^{N-1} \frac{1}{(T-t_k)} \mathbb{E}\left[ \mathrm{Tr}(\boldsymbol{\Sigma}_{T-t_k}) \right].$$

We then have $\mathbb{E}\left[ \mathrm{Tr}(\boldsymbol{\Sigma}_t) \right] = \mathbb{E}\left[ \mathbb{E}\left[ \|X_0\|^2 | X_t \right] - \|\mathbb{E}\left[ X_0 | X_t \right] \|^2 \right] \leq \mathbb{E}\left[ \|X_0\|^2 \right] = d$ for $t \in [1, T]$ and

$$\begin{aligned} \mathbb{E}\left[ \mathrm{Tr}(\boldsymbol{\Sigma}_t) \right] &= e^{2t} \mathbb{E}\left[ \mathrm{Tr}(\mathrm{Cov}(X_0 e^{-t} - X_t | X_t)) \right] \\ &= e^{2t} \mathbb{E}\left[ \mathbb{E}\left[ \|X_0 e^{-t} - X_t\|^2 | X_t \right] - \|\mathbb{E}\left[ X_0 e^{-t} - X_t | X_t \right] \|^2 \right] \\ &\leq e^{2t} \mathbb{E}\left[ \|X_0 e^{-t} - X_t\|^2 \right] \\ &\leq d e^{2t} \sigma_t^2 \\ &\leq 16 d t \end{aligned}$$

for $t \in (0, 1]$. Putting this together, we conclude that

$$\sum_{k=0}^{N-1} \int_{t_k}^{t_{k+1}} \left( \int_{t_k}^t E_{t_k,s}^{(2)} \mathrm{d}s \right) \mathrm{d}t \leq 4\kappa \mathbb{E}\left[ \mathrm{Tr}(\boldsymbol{\Sigma}_T) \right] + 4\kappa \mathbb{E}\left[ \mathrm{Tr}(\boldsymbol{\Sigma}_1) \right]$$

$$+ 4\kappa^2 \sum_{k=M+1}^{N-1} \frac{1}{(T-t_k)} \mathbb{E}\left[ \mathrm{Tr}(\boldsymbol{\Sigma}_{T-t_k}) \right]$$

$$\lesssim \kappa d + \kappa^2 d N.$$

This proves completes the proof of (16). $\qquad\square$

## F    APPLICATION OF GIRSANOV'S THEOREM

We now recall the proof of Proposition 3. The following approximation argument is essentially identical to that of Chen et al. (2023d, Section 5.2) and we reproduce it here simply for clarity of presentation.

The main ingredient in the proof will be Girsanov's theorem, which we recall below. The version we state can be obtained from Theorem 4.13 combined with Theorem 5.22 and Pages 136–139 in Le Gall (2016).

**Proposition 6** (Girsanov's Theorem). *Suppose that $(\Omega, \mathcal{F}, (\mathcal{F}_t)_{t \geq 0}, Q)$ is a filtered probability space and $(b_t)_{t \in [0,T]}$ is an adapted process on this space such that $\mathbb{E}_Q\left[ \int_0^T \|b_s\|^2 \mathrm{d}s \right] < \infty$. Let $(B_t)_{t \geq 0}$ be a $Q$-Brownian motion and define $\mathcal{L}_t = \int_0^t b_s \mathrm{d}B_s$. Then, $\mathcal{L}$ is a square-integrable $Q$-martingale. Moreover, if we define*

$$\mathcal{E}(\mathcal{L})_t = \exp\left\{ \int_0^t b_s \mathrm{d}B_s - \frac{1}{2} \int_0^t \|b_s\|^2 \mathrm{d}s \right\}$$

*for $t \in [0, T]$ and suppose that $\mathbb{E}_Q\left[ \mathcal{E}(\mathcal{L})_T \right] = 1$ then $\mathcal{E}(\mathcal{L})$ is a $Q$-martingale and so we may define a new measure $P = \mathcal{E}(\mathcal{L})_T Q$. Then, the process*

$$\beta_t = B_t - \int_0^t b_s \mathrm{d}s$$

*is a $P$-Brownian motion.*

*Proof of Proposition 3.* We will apply Girsanov's theorem on the interval $[0, t_N]$ in the case where $Q$ is the path measure of the solution to (2), $(B'_t)_{t \geq 0}$ is our $Q$-Brownian motion, and

$$b_t = \sqrt{2} \left\{ s_\theta(Y_{t_k}, T - t_k) - \nabla \log q_{T-t}(Y_t) \right\}$$

for $t \in [t_k, t_{k+1}]$ and each $k = 0, \ldots, N-1$. Note that $(b_t)_{t \in [0, t_N]}$ is an adapted process and we have

$$\mathbb{E}_Q \left[ \int_0^{t_N} \|b_s\|^2 \mathrm{d}s \right] = 2 \sum_{k=0}^{N-1} \int_{t_k}^{t_{k+1}} \mathbb{E}_Q \left[ \|\nabla \log q_{T-t}(Y_t) - s_\theta(Y_{t_k}, T - t_k)\|^2 \right] \mathrm{d}t < \infty$$

by the assumptions of Proposition 3. Therefore, if we define $\mathcal{L}_t = \int_0^t b_s \mathrm{d}B'_s$ as in Proposition 6 then $(\mathcal{E}(\mathcal{L})_t)_{t \in [0,T]}$ is a continuous local martingale (Le Gall, 2016, Proposition 5.11). So, we can find an increasing sequence of stopping times $(T_n)_{n \geq 1}$ such that $T_n \to t_N$ almost surely and $(\mathcal{E}(\mathcal{L})_{t \wedge T_n})_{t \in [0, t_N]}$ is a continuous martingale for each $n$.

Let us define $\mathcal{L}_t^n = \int_0^t b_s \mathbb{1}_{[0, T_n]}(s) \mathrm{d}B'_s$ for all $t \in [0, t_N]$ and $n \geq 1$. Then we have $\mathcal{E}(\mathcal{L})_{t \wedge T_n} = \mathcal{E}(\mathcal{L}^n)_t$, so $\mathcal{E}(\mathcal{L}^n)$ is a continuous martingale, and it follows that $\mathbb{E}_Q[\mathcal{E}(\mathcal{L}^n)_{t_N}] = 1$. Therefore, we may apply Girsanov's theorem (Proposition 6) to $\mathcal{L}^n$ on the interval $[0, t_N]$. We deduce that we may define a new probability measure $P^n := \mathcal{E}(\mathcal{L}^n)_{t_N} Q$ and a new process

$$\beta_t^n = B'_t - \int_0^t b_s \mathbb{1}_{[0, T_n]}(s) \mathrm{d}s$$

such that $(\beta_t^n)_{t \in [0, t_N]}$ is a $P^n$-Brownian motion.

Since (2) holds almost surely under $Q$, we see that

$$\mathrm{d}Y_t = \{Y_t + 2 s_\theta(Y_{t_k}, T - t_k)\} \mathbb{1}_{[0, T_n]}(t) \mathrm{d}t + \{Y_t + 2\nabla \log q_{T-t}(Y_t)\} \mathbb{1}_{[T_n, t_N]}(t) \mathrm{d}t + \sqrt{2} \mathrm{d}\beta_t^n.$$

In addition,

$$\begin{aligned}
\mathrm{KL}(Q \| P^n) &= \mathbb{E}_Q \left[ \log \frac{\mathrm{d}Q}{\mathrm{d}P^n} \right] = -\mathbb{E}_Q \left[ \log \mathcal{E}(\mathcal{L}^n)_{t_N} \right] \\
&= \mathbb{E}_Q \left[ -\mathcal{L}_{T_n} + \frac{1}{2} \int_0^{T_n} \|b_s\|^2 \mathrm{d}s \right] \\
&\leq \sum_{k=0}^{N-1} \int_{t_k}^{t_{k+1}} \mathbb{E}_Q \left[ \|\nabla \log q_{T-t}(Y_t) - s_\theta(Y_{t_k}, T - t_k)\|^2 \right] \mathrm{d}t,
\end{aligned} \tag{21}$$

since $\mathcal{L}$ is a $Q$-martingale.

Now, we consider coupling $P^n$ for each $n$ and $P^{q_T}$ by taking a fixed probability space and a single Brownian motion $(W_t)_{t \geq 0}$ on that space and defining the processes $(Y_t^n)_{t \in [0, t_N]}$ and $(Y_t)_{t \in [0, t_N]}$ via

$$\mathrm{d}Y_t^n = \{Y_t^n + 2 s_\theta(Y_{t_k}^n, T - t_k)\} \mathbb{1}_{[0, T_n]}(t) \mathrm{d}t + \{Y_t^n + 2\nabla \log q_{T-t}(Y_t^n)\} \mathbb{1}_{[T_n, t_N]}(t) \mathrm{d}t + \sqrt{2} \mathrm{d}W_t$$

and

$$\mathrm{d}Y_t = \{Y_t + 2 s_\theta(Y_{t_k}, T - t_k)\} \mathrm{d}t + \sqrt{2} \mathrm{d}W_t,$$

and taking $X_0 \sim q_T$ and setting $X_0^n = X_0$ for each $n$. Then, the law of $Y^n$ is $P^n$ for each $n$ and the law of $Y$ is $P^{q_T}$.

Fix $\varepsilon > 0$ and define $\pi_\varepsilon : \mathcal{C}([0, t_N]; \mathbb{R}^d) \to \mathcal{C}([0, t_N]; \mathbb{R}^d)$ by $\pi_\varepsilon(\omega)(t) = \omega(t \wedge (t_N - \varepsilon))$ for $t \in [0, t_N]$. Then, $\pi_\varepsilon(Y^n) \to \pi_\varepsilon(Y)$ uniformly over $[0, t_N]$ almost surely and hence $(\pi_\varepsilon)_\# P^n \to (\pi_\varepsilon)_\# P^{q_T}$ weakly (Chen et al., 2023d, Lemma 12). We then have that

$$\begin{aligned}
\mathrm{KL}((\pi_\varepsilon)_\# Q \| (\pi_\varepsilon)_\# P^{q_T}) &\leq \liminf_{n \to \infty} \mathrm{KL}((\pi_\varepsilon)_\# Q \| (\pi_\varepsilon)_\# P^n) \\
&\leq \liminf_{n \to \infty} \mathrm{KL}(Q \| P^n) \\
&\leq \sum_{k=0}^{N-1} \int_{t_k}^{t_{k+1}} \mathbb{E}_Q \left[ \|\nabla \log q_{T-t}(Y_t) - s_\theta(Y_{t_k}, T - t_k)\|^2 \right] \mathrm{d}t,
\end{aligned} \tag{22}$$

where in the first line we have used the lower semicontinuity of the KL divergence (Ambrosio et al., 2005, Lemma 9.4.3), in the second line we have used the data processing inequality, and in the third line we have used (21).

Finally, letting $\varepsilon \to 0$ we have that $\pi_\varepsilon(\omega) \to \omega$ uniformly on $[0, T]$ (Chen et al., 2023d, Lemma 13), and hence $\mathrm{KL}((\pi_\varepsilon)_\# Q || (\pi_\varepsilon)_\# P^{q_T}) \to \mathrm{KL}(Q || P^{q_T})$ (Ambrosio et al., 2005, Corollary 9.4.6). We therefore conclude by taking $\varepsilon \to 0$ in (22). $\qquad\square$

## G  Convergence of Forward Process

We show that the forward OU process converges exponentially in KL divergence under Assumption 2. We note that the exponential convergence of the OU process under various metrics is well-established, and the particular result we prove here is very similar to Lemma 9 in Chen et al. (2023a).

*Proof of Proposition 4.* Since $q_{t|0}(\mathbf{x}_t | \mathbf{x}_0) = \mathcal{N}(\mathbf{x}_t; \mathbf{x}_0 e^{-t}, \sigma_t^2 I_d)$, we have

$$\mathrm{KL}(q_{t|0}(\,\cdot\,|\mathbf{x}_0) || \pi_d) = \frac{1}{2} \left\{ d \log \sigma_t^{-2} - d + d\sigma_t^2 + \|e^{-t}\mathbf{x}_0\|^2 \right\}.$$

By the convexity of the KL divergence,

$$\begin{aligned}
\mathrm{KL}(q_T || \pi_d) &= \mathrm{KL}\Big( \int_{\mathbb{R}^d} q_{T|0}(\,\cdot\,|\mathbf{x}_0) p_{\text{data}}(\mathrm{d}\mathbf{x}_0) \Big\| \pi_d \Big) \\
&\leq \int_{\mathbb{R}^d} \mathrm{KL}(q_{T|0}(\,\cdot\,|\mathbf{x}_0) || \pi_d) p_{\text{data}}(\mathrm{d}\mathbf{x}_0) \\
&= \frac{1}{2} \left\{ d \log \sigma_T^{-2} - d + d\sigma_T^2 + e^{-2T} \mathbb{E}_{p_{\text{data}}} \left[ \|X_0\|^2 \right] \right\} \\
&= -d \log(1 - e^{-2T}) \\
&\lesssim d e^{-2T}
\end{aligned}$$

for $T \geq 1$, where we have used that $\mathbb{E}_{p_{\text{data}}} \left[ \|X_0\|^2 \right] = d$ since $\mathrm{Cov}(p_{\text{data}}) = I_d$. $\qquad\square$

## H  Linear Dependence on Data Dimension is Optimal

Suppose that we have a data distribution $p_*$ on $\mathbb{R}^d$ such that a diffusion model approximating $p_*$ using the exact scores, initialized from $q_T$ rather than $\pi_d$, and using a given sequence of discretization times $t_0, \dots, t_N$ has a KL error of $\varepsilon_*^2$. Then, if we consider approximating the product measure $p_*^{\otimes m}$ on $(\mathbb{R}^d)^m$, again using the exact score, initializing from $q_T^{\otimes m}$, and using the same sequence of discretization times, the reverse process will factorize across the $m$ copies of $\mathbb{R}^d$. Consequently, the total KL error will be $m\varepsilon_*^2$ by tensorization of the KL divergence. Thus the KL error of the diffusion model scales linearly in the data dimension in this case, demonstrating that the linear dependence of our KL bounds on $d$ is optimal.

