# OpenReview forum: "Nearly $d$-Linear Convergence Bounds for Diffusion Models via Stochastic Localization"
_ICLR.cc/2024/Conference — ICLR 2024 spotlight_

### Official Review · Reviewer_wcsw · 2023-10-22

**Soundness:** 4 excellent
**Presentation:** 3 good
**Contribution:** 4 excellent
**Rating:** 8
**Confidence:** 3

**Summary:**

The manuscript studies denoising diffusion models and specifically their convergence in KL to the "ground truth" distribution in the absence of smoothness assumptions on the ground truth distribution. The authors revisit the analysis of Chen et al. 2023a/b and improve upon it in terms of dimensional dependence (from $d^2$ to $d$, which is further argued to now be optimal). This improvement stems from a refined discretization analysis using ideas from the stochastic localization literature.

**Strengths:**

- The paper offers a rather satisfactory improvement on previous analyses of denoising by shaving of a dimensional factor (or alternatively removing extraneous assumptions such as bdd/lip etc.).

- The problem is quite clearly of interest to the community and the solution offered is also quite elegant in the way it draws on the stochastic localization literature.

- Overall, the paper is well-structured and the quality of writing is generally good. There are some caveats here that I detail below but this is nothing that cannot be adressed by a round of further proof-reading. Although I did not check the appendix in sufficient detail to vouch for correctness, I think their math is generally OK to follow.

- In terms of originality, the authors observe that Conforti et al. have produced a similar result in a parallell submission. I don't see this as an issue for publishing this manuscript at ICLR. I think it is safe to say that these have been developed in parallell and given the different techniques used they are both independently of each other of interest to the community---in my view the ideas herein are fairly original and interesting.

**Weaknesses:**

My feeling is that this is a solid piece of technical work and I did not find many weaknesses. I do have a few remarks on writing though:

- The notation is a little messy throughout and feels somewhat rushed/needs further proofreading. There are quite a few undefined/implicitly defined objects. Let me give a few examples:

    -  Previous to stating the main theorem, the main object of interest $p_{t_N}$ has never been defined.
    - The use of bold $\mathbf{x}$  for realization when defining conditional objects is a little confusing when at the same time $\mathbf{x}_*$ is a random variable. This is also in conflict with the use in equation 3 where $\mathbf{x}_t$ is a random variable drawn from $q_t$. It would be helpful if there was a consistent distinction between realization and random object throughout and if this were defined early in the manuscript.
   -  The notation for conditional expectation given the observation (function $\mathbf{a}$)  following eqn 6 is unnecessarily dense. In particular, that the LHS depends on $U_s$ whereas the RHS depends on $\mu_s$ is a little confusing at first glance. It would be good if the authors were a little more parsimonious and consistent in their notation---at least when introducing objects for the first time.


*Other comments*:


- Following Lemma 1 it would not hurt to expound a little on the significance of the covariance matrix and why it will be useful in the sequel. It would be nice if some version of what is stated "inside" the proof of thm 1, second paragraph, appears near this lemma.

- Above equation (9) there is an extraneous square in the definition of the Frobenius norm (and the square root should probably be outside of the trace)

**Questions:**

With regard to the second of part of assumption 2---you mention that your analysis is not dependent on "it"---I have clarifying question: would the result incur extra "condition numbers" or does it still work even if this covariance matrix has 0 or near 0 eigenvalues?

Could you expand on the steps taken in equation (9)? I do not immediately see how this follows from the standard Ito Isometry relating squares of stochastic integrals and integrals of their squares.

Following the statement of proposition 4 could you be a little more precise/remind us about which stochastic transformation is used in the invokation of the data-processing inequality? It seems like it is here the early stopping idea is used but I find it hard to follow.

---

> ### Author Response · Authors · 2023-11-22
>
> We thank the reviewer for their positive and detailed comments on our work. Regarding the reviewer's remarks on the writing and notation, we acknowledge the points of confusion and apologize for the lack of clarity in the original version. We will take on board the suggestions regarding the definitions of $p_{t_N}$, $\mathbf{x}$, $\mathbf{x}_\ast$, and $\mathbf{a} (U_s)$ and update the notation in the camera-ready version. We will provide an updated manuiscript reflecting the changes below by tomorrow.
>
> > Following Lemma 1 it would not hurt to expound a little on the significance of the covariance matrix and why it will be useful in the sequel.
>
> The covariance matrix $\mathbf{\Sigma_t}$ is significant since it can be related to the Jacobian of the score function (see Lemma 4). Thus, controlling $\mathbf{\Sigma_t}$ as in Lemma 1 allows us to control the Jacobian of the score function. In turn, this allows us to control time-discretization terms of the form $E_{s,t} := \mathbb{E}[||\nabla \log q_{T-t}(Y_t) - \nabla \log q_{T-s}(Y_s)||^2]$ via Ito calculus (see Lemmas 2 and 3). We will include a discussion of this intuition for Lemma 1, coming from the proof of Theorem 1, after the statement of Lemma 1 in the camera-ready version.
>
> > Above equation (9) there is an extraneous square in the definition of the Frobenius norm (and the square root should probably be outside of the trace)
>
> Thank you for spotting this; we will correct this in the camera-ready version.
>
> > With regard to the second of part of assumption 2---you mention that your analysis is not dependent on "it"---I have clarifying question: would the result incur extra "condition numbers" or does it still work even if this covariance matrix has 0 or near 0 eigenvalues?
>
> No, we do not incur extra condition numbers, and the result still holds if the covariance matrix has zero or near zero eigenvalues. For an arbitrary covariance matrix, our bounds may occur a linear dependence on $M_2 = \mathbb{E}[||x_0||^2]$ instead of $d$, but are essentially very similar in spirit. For a more detailed analysis of what exactly must be changed in our proofs and the form of the resulting bound, see our response to Reviewer 73Lh's point number 3.
>
> > Could you expand on the steps taken in equation (9)? I do not immediately see how this follows from the standard Ito Isometry relating squares of stochastic integrals and integrals of their squares.
>
> Of course; our additional exposition was probably too brief. First, writing $\textup{d} (e^t \nabla \log q_{T-t}(Y_t)) = \sqrt{2} e^t \nabla^2 \log q_{T-t}(Y_t) \cdot \textup{d} B_t'$ in integral form we get
> \begin{equation*}
>     e^t \nabla \log q_{T-t}(Y_t) - e^s \nabla \log q_{T-s}(Y_s) = \int_s^t  \sqrt{2} e^r \nabla^2 \log q_{T-r}(Y_r) \cdot \textup{d} B_r'
> \end{equation*}
> By the standard form of Ito's isometry applied to this integral, we get (under suitable integrability conditions)
> \begin{align*}
>     \mathbb{E}\Big[\Big|\Big| \int_s^t  \sqrt{2} e^r \nabla^2 \log q_{T-r}(Y_r) \cdot \textup{d} B_r' \Big|\Big|^2 \Big] & = \mathbb{E}\Big[\int_s^t (\sqrt{2} e^r \nabla^2 \log q_{T-r}(Y_r) \cdot \textup{d} B_r')^T(\sqrt{2} e^r \nabla^2 \log q_{T-r}(Y_r) \cdot \textup{d} B_r') \Big] \\\\
>     & = \mathbb{E}\Big[\int_s^t 2 e^{2r} || \nabla^2 \log q_{T-r} (Y_r)||^2_F \textup{d} r \Big]
> \end{align*}
> Combining both expressions and exchanging the order of itnegration on the RHS, we have
> \begin{equation*}
>     \mathbb{E}[||e^t \nabla \log q_{T-t}(Y_t) - e^s \nabla \log q_{T-s}(Y_s)||^2] = 2 e^{2r} \int_s^t \mathbb{E}[|| \nabla^2 \log q_{T-r} (Y_r)||^2_F] \textup{d} r.
> \end{equation*}
> Equation (9) then follows by differentiating with respect to $t$.
>
> > Following the statement of proposition 4 could you be a little more precise/remind us about which stochastic transformation is used in the invokation of the data-processing inequality? It seems like it is here the early stopping idea is used but I find it hard to follow.
>
> We are applying the data processing inequality with the transformation $f : (\omega_t)  \mapsto \omega_{t_N}$ which takes a path $(\omega_t) \in C([0,t_N], \mathbb{R})$ on $[0,t_N]$ and returns its endpoint. Under this transformation, $Q$ and $P^{\pi_d}$ are mapped to $q_\delta$ and $p_{t_N}$ respectively. (This is a similar argument to that used in Chen et al. (2023d).)

---

> > ### Comment · Reviewer_wcsw · 2023-11-22
> >
> > I don't have much to add at this point except that I'd like to thank the authors for an informative response to my questions. All questions I had have been answered and I think the changes indicated by the authors will make the exposition a little clearer.
> >
> > I'd like to keep my vote for acceptance.

---

### Official Review · Reviewer_73Lh · 2023-10-28

**Soundness:** 3 good
**Presentation:** 3 good
**Contribution:** 3 good
**Rating:** 8
**Confidence:** 4

**Summary:**

This paper studies sampling from a data distribution on R^d using denoising diffusion.
The main result is an improved convergence rate under only a second moment assumption on the data distribution.
This bound improves the dependence of rate on d from quadratic in Chen et al. to nearly linear.
Other parameter dependencies are the same as in Chen et al.

**Strengths:**

This paper is pretty well-written.
The punchline is clearly addressed within the first 3 pages.
The key technique that makes the rate improvement possible is a differential inequality for one of the error terms, instead of using Lipschitz bounds which apparently require additional model assumptions.
I particularly appreciate it that the authors make it clear what is novel in their contribution and what has been well-established in the literature.
In the latter case, proper references are given.

**Weaknesses:**

I don't see obvious technical weaknesses.
Here are two suggestions.

1. Though this is a purely theoretical paper, given how successful diffusion model is in practice, it might add some value to show some experimental results even for very simple data distribution.

2. Several auxiliary results are known in the literature or can be adapted from known results in a straightforward way. However, proofs for many of these results are still given in the appendix. I understand the intention to increase readability. However, according to my personal taste it feels that there's some room to streamline the proof and make the paper more succinct. To clarify, this is a matter of presentation and either way does not affect my evaluation of the result.

**Questions:**

1. I encourage the authors to add "nearly linear" in the title. This is a standard terminology (at least in theoretical computer science) for hiding log factors.

2. One dumb question: doesn't the second part of Assumption 2 imply its first part? Or am I mis-interpreting the meaning of second moment of a distribution on R^d?

3. It is claimed that the isotropic condition in Assumption 2 is not essential. However, this has never been formally justified, if I'm not missing it. Which nuisance of the proof needs to be changed if the covariance is general? In fact, I have some doubt on the claim that this assumption is immaterial. What if the covariance is not strictly positive definite? What if the covariance is **unknown**?

4. In the line following equation (9), there seems to be some typos in the equations for verifying the applicability of Fubini.

5. Right before the statement of Lemma 6, it's claimed that the existence of t_M = T-1 is WLOG. This is likely technical but I suggest briefly justify it. I guess if this does not hold, a tiny approximation error needs to be introduced.

6. Please introduce the notation $\lesssim$ somewhere if it's not been done yet. I assume that this means the LHS is upper bounded by the RHS multiplied by a numerical constant.

7. It's argued informally in section 4 (with some details in appendix G) that the linear dependence of the convergence rate on d is necessary.
This seems like a nice observation. Is it possible to turn it into a formal proposition and specify how this lower bound depends on other parameters besides d? In fact, this observation is a little surprising to me since for product distribution, the problem is effectively 1-dimensional however denoising diffusion still incurs a dimension-dependence. Could the authors offer some more intuition? Thanks in advance.

---

> ### Author Response · Authors · 2023-11-22
>
> We thank the reviewer for their detailed engagement with our work and thoughtful comments and suggestions. In response to the questions raised by the reviewer:
>
> 1. We agree this is a sensible suggestion, and will update the title to "Nearly $d$-Linear Convergence Bounds for Diffusion Models via Stochastic Localization".
>
> 2. Yes, the first part of Assumption 2 is technically superfluous. We chose to state it in this way to separate the assumption that the data distribution has finite second moments (which is absolutely required by our methods) and the assumption that $p_{\textup{data}}$ is isotropic (which can be weakened, see next comment).
>
> 3. The main changes required are in the proofs of Lemmas 5 and 6 and Proposition 4. Suppose that instead of assuming that $\textup{Cov} (p_{\textup{data}}) = I_d$, we only assume that we have finite second moments and let $M_2 = \mathbb{E}[||x_0||^2]$. Then, Lemmas 2, 3 and 4 hold unchanged. Also, the proof of (13) can be adapted, replacing each time we use $ \mathbb{E}[||x_0||^2] = d$ with $ \mathbb{E}[||x_0||^2] = M_2$; we find that all instances of $M_2$ cancel in the final result and so Lemma 5 also holds unchanged. The only change in the proof of Lemma 6 comes at the top of page 18, where we find instead that $\mathbb{E}[\textup{Tr}(\mathbf{\Sigma_t})] \leq M_2$ for $t \in [1,T]$. We deduce that Lemma 6 holds almost unchanged, except the RHS of (17) should contain an additional term of $\kappa M_2$. The final change that must be made is in Proposition 4, where the bound becomes $\lesssim (d + M_2) e^{-2T}$ (this can be easily seen from the proof). Putting this all together, we end up with the final result that $\textup{KL}(q_{\delta}||p_{t_N}) \lesssim \varepsilon^2_{\textup{score}} + \kappa^2 d N + \kappa d T + \kappa M_2 + (d + M_2) e^{-2T}.$ This bound holds even in the case where the covariance is not strictly positive definite, and in the case that it is unknown in advance. Since we expect $M_2$ to scale linearly in $d$, we consider this result to be in essentially the same spirit as Theorem 1, with only minor adaptations required.
>
> 4. Thank you for drawing our attention to this line. In fact, in the first version of the manuscript, we made a mistake when giving the relevant inequalities which permit the application of Fubini. As can be seen from writing out the application of Ito's isometry in full (see our response to Reviewer wcsw below), the correct justification should be $\int_s^t e^{2r} \mathbb{E}[||\nabla^2 \log q_{T-r}(Y_r)||._F^2] \textup{d} r < \infty$ (which follows from (13), as stated above equation (9)). We will correct this in the camera-ready version.
>
> 5. Indeed this is merely a technicality, and our justification is as follows. If we wanted to work in full generality, we would instead define $M$ to be an index such that $t_M \in [T-2, T-1]$, noting that this must exist since the step size $\gamma_k$ satisfies $\gamma_k \leq \kappa \leq 1$ (for small enough $\kappa$). Then, we would divide our time period into the intervals $[0,t_M]$ and $[t_M, T-\delta]$ and treat them separately. The interval $[0,t_M]$ is a subset of $[0,T-1]$ and so could be treated exactly as before. We would treat $[t_M, T-\delta]$ the same way as $[T-1, T-\delta]$ with the only change being that for $t \in [t_M, T-\delta]$ we now have $T-t \leq 2$ rather than $T-t \leq 1$ and so we would end up with different constant factors in our upper and lower bounds on $\sigma_{T-s}^2$ and $\gamma_k \sigma_{T-t_k}^{-4}$. These different constant factors do not affect the overall result.
>
> 6. We will include a definition in the camera-ready version.
>
> 7. Intuitively, we see the dimension dependence as arising unavoidably from the use of KL divergence to measure the distance between the true and approximate data distribution. Even if the problem factorizes completely, so that it can essentially be viewed as a product of independent 1-dimensional approximation problems, the fact that the KL divergence tensorizes across dimensions means that the error will still scale linearly in the dimension. We considered trying to formalize a lower bound on the iteration complexity in terms of $d$ and the other problem parameters. However, we came to the conclusion that it would be challenging to supply a lower bound that had realistic dependence on both the data dimension and the step size, since the two appear to somewhat trade off against each other for reasons discussed in Section 4. Getting more precise about exactly how the dependence on the dimension and the step size trade off would likely require a much more thorough theoretical investigation.

---

### Official Review · Reviewer_TDrD · 2023-10-31

**Soundness:** 4 excellent
**Presentation:** 3 good
**Contribution:** 4 excellent
**Rating:** 8
**Confidence:** 3

**Summary:**

Many current state-of-the-art generative models are denoising diffusion models. These models take samples from a data distribution, run them through a forward SDE, which progressively adds Gaussian noise to the samples, and then seek to learn the reverse SDE using neural networks. This paper seeks to prove mixing bounds for the reverse diffusion process after the denoising score model has been learned. In particular, under the assumptions of a learned score function with an L2 error bound and a data distribution with identity covariance, the authors use the theory of stochastic localization to yield a bound on the KL divergence between the generated samples and the data distribution. This bound those that the number of iterates to reach error $\epsilon$ scales linearly in $d$, quadratically in the forward time $T$, and $O(\epsilon^{-2})$ with respect to $\epsilon$.

**Strengths:**

- This work tackles a highly relevant problem on the convergence of denoising diffusion models.
- The bound given scales linearly in the dimension, which seems optimal in the worst case, as the authors point out in the appendix.
- The bound given accounts for a realistic setting, where the chain has early stopping, approximate initialization, and discrete reverse time discretization.
- The bound brings new theoretical tools of stochastic localization, which should be more broadly attractive to the community working on diffusion models.
- The authors explain the proof in detail in the paper.

**Weaknesses:**

- There are no simulations or experiments to support the results. However, this is a smaller weakness since the work is primarily theoretical.
- I wish that there was more discussion of the key lemma. Why is stochastic localization needed to prove it? How can we interpret it?
- I have some issues with the presentation of the main theorem in the paper. This is important since about a third of the paper is dedicated to explaining the proof of Theorem 1. I found their explanation to be hard to follow. First, I would appreciate a roadmap explaining precisely what steps are new/novel and what steps are guided by past work. In particular, it seems that the work on the discretization error is new, so the authors are then able to apply the bound from stochastic localization. However, is the perspective taken where they view the single process $(Y_t)$ as solutions to (2) and (4) under different measures new? The biggest issue, though, in my opinion, is that much of the proof is left to the appendix. I understand this is necessary, but without more interpretation in the paper to help the reader understand what is happening, it is easy to get lost. Improving this explanation would greatly help the paper since it is primarily theoretical. Some suggestions are the following.

Setup
- More clearly define the measures $P^{\cdot}$ and $Q$, and explain how $(Y_t)$ is a solution to (2) and (4) under these measures.
Bound on the discretization error
- An initial lemma saying the final bound for this section so that the reader can put into context the computations being done. Then, put Lemmas 2-6 into context for how they help with the final bound as they are discussed.
Girasanov
- Again, state the final bound as a lemma so that the discussion can be put into context. This can then be pointed to throughout.

**Questions:**

- While the linear dependence in the dimension is expected in the worst case, does the analysis of this paper and similar ones point toward dimension-independent rates under further assumptions on the score? Or is this a limitation of the Langevin diffusion? Could other processes help to alleviate this?

---

> ### Author Response · Authors · 2023-11-22
> **Part 1**
>
> We thank the reviewer for their detailed comments on our work. We agree with the reviewer that our contributions are primarily theoretical. We also take on board their comments regarding the presentation of our main Theorem.
>
> > I wish that there was more discussion of the key lemma. Why is stochastic localization needed to prove it? How can we interpret it?
>
> The covariance matrix $\mathbf{\Sigma_t}$ is important since it can be related to the Jacobian of the score function (see Lemma 4). Lemma 1 thus allows us to control the Jacobian of the score function, which in turn allows us to control time-discretization terms of the form $E_{s,t} := \mathbb{E}[||\nabla \log q_{T-t}(Y_t) - \nabla \log q_{T-s}(Y_s)||^2]$ via Ito calculus (see Lemmas 2 and 3).
>
> Lemma 1 could be proved without reference to stochastic localization. The proofs of Propositions 1 and 2 boil down to standard applications of Ito calculus, and these could be rephrased entirely in the language of diffusion models. However, we choose to present Lemma 1 in the context of stochastic localization in order to provide due credit, since this was the context in which we discovered these ideas, and because we feel that the relationship to stochastic localization provides additional intuition and motivation for our proofs. We will include some more discussion of Lemma 1 after the statement in the camera-ready version (see also our reply to Reviewer wcsw).

---

> ### Author Response · Authors · 2023-11-22
> **Part 2**
>
> > I have some issues with the presentation of the main theorem in the paper. This is important since about a third of the paper is dedicated to explaining the proof of Theorem 1. I found their explanation to be hard to follow. First, I would appreciate a roadmap explaining precisely what steps are new/novel and what steps are guided by past work. In particular, it seems that the work on the discretization error is new, so the authors are then able to apply the bound from stochastic localization.
>
> We apologize for the lack of clarity surrounding this overarching structure in the original manuscript. We will provide an updated manuscript by tomorrow with a clearer roadmap at the start of Section 3 explaining which sections of our work are novel. We will also slightly modify the structure of Section 3 to make it easier to get an overview of the proof at a glance, including adding in lemmas summarising the key results of the sections on the bound on the discretization error and the application of Girsanov's theorem.
>
> With the updated structure, our proof comes in three key steps. We view Step 1 as our main novel contribution, while Steps 2 and 3 closely follow Chen et al. (2023a;d).
>
> **Step 1:** We bound the error from discretizing the reverse SDE. Previous works bound $E_{s,t} := \mathbb{E}[||\nabla \log q_{T-t}(Y_t) - \nabla \log q_{T-s}(Y_s)||^2]$ using a Lipschitz assumption. We use a new Ito calculus argument to get a differential inequality for $E_{s,t}$ (Lemmas 2 and 3), relate the coefficients of this inequality to $\mathbf{m}_t$ and $\mathbf{\Sigma}_t$ using known results (Lemma 4), and bound the differential inequality coefficients using our key Lemma 1 (resulting in Lemmas 5 and 6).
>
> **Step 2:** We bound the KL distance between the path measures of the true and approximate reverse process. We use an analogous Girsanov-based method to Chen et al. (2023a;d).
>
> **Step 3:** We use the data processing inequality to bound $\textup{KL}(q_{\delta}||p_{t_N})$ in terms of the distance between reverse path measures and the distance between $q_T$ and $\pi_d$. We bound the former using Step 2 and the latter using the convergence of the OU process (Proposition 4), as in Chen et al. (2023a;d).
>
> > However, is the perspective taken where they view the single process as solutions to (2) and (4) under different measures new?
>
> No, this is also done in Chen et al. (2023a;d). We will make this clear in our updated introduction to Section 3.
>
> > The biggest issue, though, in my opinion, is that much of the proof is left to the appendix. I understand this is necessary, but without more interpretation in the paper to help the reader understand what is happening, it is easy to get lost. Improving this explanation would greatly help the paper since it is primarily theoretical. Some suggestions are the following...
>
> Hopefully, our improved introduction to Section 3 will help to mitigate any lack of clarity caused by the fact that some technical details must inevitably be left to the appendix. In particular, we have taken on board the reviewer's helpful suggestions to pull out the key results from the sections on the bound on the discretization error and the application of Girsanov's theorems, and will use these to add more context to the proofs.
>
> > While the linear dependence in the dimension is expected in the worst case, does the analysis of this paper and similar ones point toward dimension-independent rates under further assumptions on the score? Or is this a limitation of the Langevin diffusion? Could other processes help to alleviate this?
>
> Even for smooth data distributions such as Gaussians, we expect that the convergence rate measured in KL divergence will depend linearly on the data dimension. This is unavoidable since the KL divergence tensorizes over independent dimensions, so we will inevitably pick up a linear scaling in $d$ in the time-discretization error (unless the error is precisely zero, which will not happen if we include some time-discretization). Even the Langevin diffusion with respect to a strongly log-concave distribution (or more generally a distribution satisfying a log-Sobolev inequality) will pick-up a linear dimension dependence once discretised.

---

> > ### Comment · Reviewer_TDrD · 2023-11-22
> >
> > I appreciate your responses to my feedback, and since it addresses my main concerns, I will raise my score and advocate for acceptance.

---

### Official Review · Reviewer_4GBU · 2023-10-31

**Soundness:** 4 excellent
**Presentation:** 3 good
**Contribution:** 3 good
**Rating:** 6
**Confidence:** 4

**Summary:**

The paper gives a new sampling error bound for diffusion models. Roughly speaking, a diffusion model is a method for sampling from a data distribution $q_0$ that "works" whenever one can approximate the scores $\nabla q_t(\cdot)$ well, where $q_t$ is the density of a particle undergoing Ornstein-Uhlenbeck dynamics from $q_0$. Usually, score approximations would be found with deep neural networks.

Previous results by other authors obtained bounds relating the quality of score approximations to the distance between $q_0$ and the distribution output by the diffusion model. The present paper gives improved bounds: the main difference is that the error in approximating scores is not multiplied by any time- or -dimension dependent factors, but only by a universal constant.   The proof of this result follows the outline of previous work by Chen, Lee and Lu; the main modification (according to the authors) is in dealing with discretization errors.is

**Strengths:**

The bound is an improvement over previous results that is quite natural. Proof techniques are elegant.

**Weaknesses:**

The main proof outline is not too novel. The condition on the final step sizes is not very natural.

**Questions:**

Can you give me a better sense on where the restrictions on the discretization come from?

---

> ### Author Response · Authors · 2023-11-22
>
> We thank the reviewer for their engagement with our work. Regarding the novelty of our work, we have provided a detailed discussion explaining which parts of our work are novel and which are related to the previous literature in response to Reviewer TDrD below.
>
> > Can you give me a better sense on where the restrictions on the discretization come from?
>
> Our intuition for the condition on the step sizes is as follows. Consider a diffusion model with an OU noising process where the data distribution is a point mass at the origin and we have learned to perfectly approximate the scores of the marginal distributions. Then, it can be shown that the KL distance between the true and approximate reverse path measures will be (roughly) $\sum_k \gamma_k^2/ (T-t_{k+1})^2$, assuming we initialize the reverse process at $q_T$. Hence, we should choose our discretization scheme to ensure that this quantity is as small as possible. One intuitive way to make this quantity reasonably small is to make all of the ratios $\gamma_k/ (T-t_{k+1})$ equal to some constant $\kappa$, and then choose the smallest $\kappa$ such that this is possible. This will force the step sizes to decay exponentially as we approach the starting time.
>
> The condition $\gamma_k \leq \kappa(T-t_{k+1})$ can then be thought of as forcing at least exponential decay in the step sizes near the original starting time. Note that smaller step sizes nearer the starting time are beneficial, because for non-smooth data distributions the scores will explode as $t \rightarrow 0$ and so we should take smaller step sizes to compensate for the decreased regularity. The condition $\gamma_k \leq \kappa$ is included to ensure that in addition to decaying exponentially, the step sizes also all uniformly converge to zero at a rate controlled by $\kappa$.

---

### Official Review · Reviewer_GLNq · 2023-11-01

**Soundness:** 3 good
**Presentation:** 4 excellent
**Contribution:** 2 fair
**Rating:** 6
**Confidence:** 4

**Summary:**

The paper improves the analysis for diffusion models, primarily in the setting without uniform smoothness of $\log p_t$. In this setting, it improves the dependence on time discretization (under an $\epsilon^2$ close score matching estimator) from $d^2$ to $d$, by analogy with techniques in stochastic localization.

**Strengths:**

The tight rate in dimension is genuinely novel, and the setting where $\log p_t$ is not uniformly smooth is arguably the more important regime.

The incorporation of stochastic localization techniques, seen also in Montanari (2023) and other works, is quite elegant and possibly suggests additional improvements. See questions.

**Weaknesses:**

The main contributions of this work are somewhat narrow, found primarily Lemmas 2, 4, 5, 6, which in my opinion are algebraic consequences of insights developed in prior work. For instance, the original perspective of diffusion models as stochastic localizers comes from Montanari, and the key lemmas used in analyzing stochastic localization are from Montanari (2023) and some of Eldan’s works. The score-matching error and OU error are handled by prior work on diffusion models, from Chen et al.

Consequently, the paper could benefit from additional results, see my questions below.

Despite this, the improvement in rate made by this paper is quite significant and definitely merits publication. Consequently, I am voting for acceptance.

**Questions:**

Have the authors attempted to apply such techniques to other families of diffusion models, beyond those using the Ornstein-Uhlenbeck forward flow? For instance those found in later work by Chen et al. [1]

[1] Chen, Sitan, Giannis Daras, and Alex Dimakis. "Restoration-degradation beyond linear diffusions: A non-asymptotic analysis for ddim-type samplers." International Conference on Machine Learning. PMLR, 2023.

Below are some minor corrections/questions:
Frobenius norm squared should be just Tr(A^\top A) (page 6)
Girsanov’s Theorem is given with the condition that $b_t$ is “previsible”, but I am not too clear on what previsible means in this context (it is not given in Le Gall). The standard statement is just that $b_t$ is an adapted process satisfying Novikov’s condition.

---

> ### Author Response · Authors · 2023-11-22
>
> Thank you for your comments on our work and suggestions for additional avenues of exploration. Regarding the novelty of our work, we have provided a detailed discussion explaining which parts of our work are novel and which are related to the previous literature in response to Reviewer TDrD below.
>
> > Have the authors attempted to apply such techniques to other families of diffusion models, beyond those using the Ornstein-Uhlenbeck forward flow? For instance those found in later work by Chen et al. [1]
>
> First, we note that our results should apply directly to any case where the drift term of the noising SDE is replaced by some linear function of $X_t$; this is since the resulting noising process can be transformed into an OU process via a combination of a linear transformation and rescaling the time variable. Consequently, an analogous version of Theorem 1 will hold in these cases. In particular, this means that our results also hold for the variance-exploding SDE, as described in Song et al. (2021b). We will add a remark to this effect in the camera-ready version.
>
> For more general noising SDEs of the type described in Chen et al. [1], our proof methods would need significant adaptation. This is because the linearity of the SDE is integral to Lemma 1 and therefore to our derivation of Lemma 5. However, if one could derive a result analogous to Lemma 5 in the more general case then this may be sufficient; the form that would be required for the rest of our argument to go through is, roughly, that $E_Q [||\nabla \log q_{T-s}(Y_s)||^2]$ and  $E_Q [|| \nabla^2 \log q_{T-t} (Y_t) ||^2]$ should both be bounded by terms of the form $A(t) + \frac{\textup{d}}{\textup{d} t} B(t)$ where $A(t) \lesssim d \sigma_{T-t}^{-4}$ and $B(t) \lesssim d \sigma_{T-t}^{-2}$. It may be possible to derive such a bound by assuming, for example, a linear upper bound on the drift coefficient and a lower bound on the diffusion coefficient, and using a comparison method with a linear SDE. However we have not explored these details further, as doing so would likely require significant additional technical work beyond the scope of the current paper.
>
> In addition, in practice non-linear SDEs are rarely used as noising processes, since the linear structure in the SDE greatly simplifies training procedures. Therefore, we have chosen to focus on linear noising SDEs in the current work.
>
> We will also incorporate the minor comments on the definition of the Frobenius norm and the statement of Girsanov's theorem in the camera-ready version.

---

### Official Review · Reviewer_sqy6 · 2023-11-01

**Soundness:** 3 good
**Presentation:** 3 good
**Contribution:** 3 good
**Rating:** 8
**Confidence:** 4

**Summary:**

The paper provides bounds on the number of steps of the discretized SDE in diffusion models required to approximate a general distribution with finite second moments upto a certain accuracy. The central novelty of the paper's analysis is the incorporation of the decay of the expected covariance of the denoising posterior measure during the process of the sampling. The rate of the decay is obtained through results in the stochastic localization literature, which is subsequently combined with the existing analysis based on Girsanov's theorem. The analysis leads to an improved dependence of the iteration-complexity on the input dimension in the absence of a uniform-Lipschitz assumption on the score function.

**Strengths:**

- Theoretical analysis of diffusion models is an active area with several open problems.
- The contribution to the proof technique, albeit straightforward, could be useful for downstream theoretical analysis of aspects such as sample and computational complexity, as well as for proofs related to algorithmic stochastic localization in spin-glass models.
- The paper is well written.
- The proofs are easy to follow and largely self-contained.
- The paper adequetely acknowledges the inspiration from the stochastic localization literature towards obtaining the bounds, even though the bounds can be obtained directly using Itô calculus. This helps the reader appreciate the connection with stochastic localization and the motivation behind the results.

**Weaknesses:**

- Majority of the proof closely follows the analysis of Chen et al. (2023a;d).
- The paper improves upon best existing bounds in the absence of uniform-Lipschitzness of the score function. However, more explanation is required to understand why Lipschitzness is a limiting assumption.  For instance, while for discrete measures it is apparent that one must resort to early stopping on a corrupted version of the distribution, it requires further justification for measures absolutely continuous w.r.t the Lebesgue measure. Futhermore, the proofs in the algorithmic stochastic localization works such as El Alaoui et al. (2022), themselves rely on the Lipschitzness of the estimated score function.
- Assumption 1 of approximating the score to arbitrary error, while utilized in existing works, is unrealistic for several setups. For instance, some recent works have highlighted statistical and computational barriers towards the estimation of the score function, in particular the works El Alaoui et al., 2022; Montanari & Wu, 2023, as well as:

   Biroli, G., & Mézard, M. (2023). Generative diffusion in very large dimensions. arXiv preprint arXiv:2306.03518.

   Ghio, D., Dandi, Y., Krzakala, F., & Zdeborová, L. (2023). Sampling with flows, diffusion and autoregressive neural networks: A spin-glass
   perspective. arXiv preprint arXiv:2308.14085.

   A discussion of the limitations of assumption 1 would strengthen the paper.

**Questions:**

## Questions
- For what classes of distributions is the uniform Lipshitzness assumption restrictive? What iteration complexity could one expect under smoothness assumptions non-uniform in time?
- In Lemma 5, the covariance bound in Lemma 1 is utilized to bound the expected Frobenius norm of the Jacobian of the score function. Can this be interpreted as establishing an average-Lipschitzness of the score function?
- Are deterministic sampling schemes expected to yield identical iteration complexities?

## Suggestions
- In the title, it should be clarified that "linear convergence" refers to being linear in the input dimension, to avoid confusion with the use of the term "linear convergence" in the optimization literature.
- To improve clarity, one could clarify that the SDE in Eq. 6 and the noisy observation process in Eq. 5 are equivalent only in law.
- I suggest mentioning in sections 1.1 and 1.2 how the posterior means are related to the score function.

---

> ### Author Response · Authors · 2023-11-22
> **Part 1**
>
> We thank the reviewer for their thoughtful remarks and suggestions.
>
> > Majority of the proof closely follows the analysis of Chen et al. (2023a;d).
>
> We agree that the structure of our proof closely resembles that of Chen et al. (2023a;d). We see our main contribution as being a more refined treatment of the time-discretization component of their bounds, and the relationship between this treatment and the stochastic localization literature. We are planning to update the introduction to Section 3 with a more thorough overview of which parts of our work are novel and which parts follow previous results; see our response to reviewer TDrD below for more details.
>
> > However, more explanation is required to understand why Lipschitzness is a limiting assumption. For instance, while for discrete measures it is apparent that one must resort to early stopping on a corrupted version of the distribution, it requires further justification for measures absolutely continuous w.r.t the Lebesgue measure.
>
> To clarify, our motivation for wishing to remove the Lipschitz assumption is two-fold. Firstly, as the reviewer comments, this assumption is inappropriate for discrete measures, but also for measures which are supported on a sub-manifold of $\mathbb{R}^d$. We expect that many data distributions of interest will be of this form.
>
> Secondly, even if the score of the data distribution is Lipschitz, the Lipschitz constant will be very large when $t \rightarrow 0$ for many data distributions (e.g. those approximately supported on a sub-manifold). Crucially, in many cases we expect that the Lipschitz constant will scale with the data dimension (as is seen, for example, in Chen et al. (2023a), where they use early stopping to ensure that the scores of the marginal distributions are uniformly Lipschitz, but in doing so pick up an extra $d$ dependence in the Lipschitz constant). As such, previous assumptions of a uniform-in-time Lipschitz constant can hide the full dimension dependence. We therefore consider them misleading in many practical cases, even if the data distribution is smooth. As this intuition is a key part of the motivation for our work, we thank the reviewer for highlighting it and will clarify the presentation in the camera-ready version.
>
> > Assumption 1 of approximating the score to arbitrary error, while utilized in existing works, is unrealistic for several setups.
>
> We agree that Assumption 1 has some important limitations. On the one hand, there are limitations on the estimation of the score function, as highlighted by the reviewer. In addition, while the results in our paper would suggest that the convergence rate of the diffusion model is determined by the error in the discretization of the reverse process, we find in practice that the score-approximation error $\varepsilon_{\textup{score}}^2$ is the dominant term in the bound of Theorem 1. Therefore, Assumption 1, which states that we may learn the score to within an arbitrary approximation error $\varepsilon_{\textup{score}}^2$, may be inappropriate in practice. We will include an additional mention of these limitations in the camera-ready version of our work.
>
> Nevertheless, Assumption 1 is the standard assumption on the error of the score approximation used in the rest of the literature on the convergence of diffusion models. We therefore choose to focus on it in our work. A more detailed analysis of its appropriateness would be somewhat orthogonal to the main thrust of the present work.
>
> > For what classes of distributions is the uniform Lipshitzness assumption restrictive? What iteration complexity could one expect under smoothness assumptions non-uniform in time?
>
> Hopefully, our response above clarifies that the uniform Lipschitzness assumption is restrictive for data supported on a sub-manifold, and also may hide sub-optimal dependence on the data dimension in many additional cases -- for example, any case where the Lipschitz constant implicitly scales with the data dimension, which may occur if the data distribution is approximately distributed on a sub-manifold. Using a Lipschitz constant which varies in time will not automatically avoid these problems; indeed, the work of Chen et al. (2023a) only assumes Lipschitz smoothness at $t=0$, but still obtains quadratic dependence in the data dimension. Fundamentally, it seems to us that any proof method which aims to control a quantity of the form $E_{s,t} := \mathbb{E}[||\nabla \log q_{T-t}(Y_t) - \nabla \log q_{T-s}(Y_s)||^2]$ using Lipschitzness of the scores - i.e. a supremum bound - rather than working in expectation will pick up an extra factor of $d$ and so be suboptimal compared to the results in our work.

---

> ### Author Response · Authors · 2023-11-22
> **Part 2**
>
> > In Lemma 5, the covariance bound in Lemma 1 is utilized to bound the expected Frobenius norm of the Jacobian of the score function. Can this be interpreted as establishing an average-Lipschitzness of the score function?
>
> Yes; the operator norm of the Jacobian of the score can be thought of as a "local Lipschitz constant" of the score function. Hence Lemma 5, which controls the expected Frobenius norm of the Jacobian, can be thought of as controlling the "expected local Lipschitzness" of the score (up to the equivalence of the Frobenius and operator norms).
>
> > Are deterministic sampling schemes expected to yield identical iteration complexities?
>
> We expect that deterministic sampling schemes will still have a linear dependence on the data dimension (as our example in Appendix G will continue to apply in this setting). However, we expect deterministic sampling schemes to generally have a better dependence on the step size. This is seen, for example, in this paper by Li et al. (2023): https://arxiv.org/pdf/2306.09251.pdf. However, convergence results for deterministic sampling typically require either additional smoothness assumptions on the data distribution (see e.g. Benton et al. (2023) https://arxiv.org/abs/2305.16860), or a small amount of stochasticity in the sampling scheme in order to smooth the reverse process (see e.g. Chen et al. (2023c) https://arxiv.org/abs/2305.11798, or Albergo et al. (2023) https://arxiv.org/abs/2303.08797).
>
> We also thank the reviewer for their additional suggestions. We will update the title of our work to ``Nearly $d$-Linear Convergence Bounds for Diffusion Models via Stochastic Localization'' and include their suggested changes on Equations 5,6 and the comment on the posterior means in the camera-ready version.

---

### Author Response · Authors · 2023-11-22
**Many thanks to reviewers, updated manuscript to follow**

Many thanks to all of the reviewers for their thorough and insightful comments on our submission. We have provideded detailed responses to individual reviewers below, and hope that these will address your concerns. In addition, we intend to submit an updated version of the manuscript reflecting the changes proposed in the dicsussions below by tomorrow.

---

> ### Author Response · Authors · 2023-11-23
> **Revised version uploaded**
>
> We have now uploaded a revised version of our manuscript reflecting the changes proposed in the discussion process.

---

### Meta-Review · Area_Chair_RT6b · 2023-12-05

**Metareview:**

This is a solid theoretical paper improving significantly on previously known bounds. It utilized the connection between diffusion and stochastic localization, put forward in previous work, but that deserves to be further highlighted in the wider ML community. A clear acceptance, all referees agree. The discussion should give the authors numerous tips on how to improve the presentation of their results.

**Justification For Why Not Higher Score:**

While the results are very nice, the proof technique does not seem to be drastically novel and the topic is extending works on related weaker bounds. I do not have enough papers to calibrate, but feel that a spotlight is the right hit here.

**Justification For Why Not Lower Score:**

The connection between diffusion and stochastic localization techniques provides powerful insight. While not novel in this paper, highlighting it further would seem appropriate.

---

### Decision · Program_Chairs · 2024-01-16

Accept (spotlight)